# External location of touch is constructed post-hoc based on limb choice

**Femke Maij[1][†], Christian Seegelke[2,3][†], W Pieter Medendorp[1], Tobias Heed[2,3]\***

[1]Donders Institute for Brain, Cognition and Behaviour, Radboud University Nijmegen, Nijmegen, Netherlands; [2]Faculty of Psychology and Sports Science, Bielefeld University, Bielefeld, Germany; [3]Center for Cognitive Interaction Technology, Bielefeld University, Bielefeld, Germany

**Abstract** When humans indicate on which hand a tactile stimulus occurred, they often err when their hands are crossed. This finding seemingly supports the view that the automatically determined touch location in external space affects limb assignment: the crossed right hand is localized in left space, and this conflict presumably provokes hand assignment errors. Here, participants judged on which hand the first of two stimuli, presented during a bimanual movement, had occurred, and then indicated its external location by a reach-to-point movement. When participants incorrectly chose the hand stimulated second, they pointed to where that hand had been at the correct, first time point, though no stimulus had occurred at that location. This behavior suggests that stimulus localization depended on hand assignment, not vice versa. It is, thus, incompatible with the notion of automatic computation of external stimulus location upon occurrence. Instead, humans construct external touch location post-hoc and on demand.

## Introduction

Spatial perception and actions rely on multiple spatial codes, often associated with different reference frames. For instance, the accuracy of pointing or reaching with an arm or finger to a visual target depends not only on the position of target relative to gaze (*Fiehler et al., 2011*; *Thompson et al., 2014*), but also on salient world-centered landmarks (*Schütz et al., 2013*). Similarly, judgment of visual location during whole-body movement is influenced by a target's position relative to gaze, as well as by the location of the target relative to the body (*Tramper and Medendorp, 2015*).

In touch, too, space is coded in several reference frames. Touch activates specialized sensory receptors embedded in the skin, and the arrangement of the peripheral sensors is reflected in the homuncular organization of primary somatosensory cortex (*Penfield and Boldrey, 1937*; *Roux et al., 2018*), referred to as a skin-based or somatotopic layout. However, because our body can take various postures, the stimulus location in space – often referred to as its external location – must be derived by combining skin location and body posture, a process termed tactile remapping (*Heed et al., 2015a*). Indeed, there is evidence that external tactile locations can be coded in a gaze-centered reference frame (*Harrar and Harris, 2010*; *Mueller and Fiehler, 2014a*; *Mueller and Fiehler, 2014b*), but also relative to anchors such as the head, torso, and hand (*Alsmith et al., 2017*; *Heed et al., 2016*).

It is less clear, however, according to which principles these different spatial codes are employed. Both bottom-up features such as the availability of sensory information (*Bernier and Grafton, 2010*) and the spatial reliability of a sensory channel (*Ernst and Banks, 2002*; *van Beers et al., 2002*), as well as top-down information such as task-constraints (*Badde et al., 2016*; *Schubert et al., 2017*), action context (*Mueller and Fiehler, 2014b*), and cognitive load (*Badde et al., 2014*) can affect the relative contributions of different reference frames, presumably in a weighted manner

**\*For correspondence:**
tobias.heed@uni-bielefeld.de

[†]These authors contributed equally to this work

**Competing interests:** The authors declare that no competing interests exist.

(*Angelaki et al., 2009*; *Atsma et al., 2016*; *Badde and Heed, 2016*; *Ernst and Di Luca, 2011*; *Kayser and Shams, 2015*; *Lohmann and Butz, 2017*; *Tramper and Medendorp, 2015*). Yet, whereas there is widespread consensus that each spatial code can have more or less influence depending on the specific situation, it is currently not known whether all putative codes are always constructed, or whether they are only computed based on demand.

For touch, it has been suggested that the construction of spatial location is an automatic process, implying that any tactile input is remapped into an external code, irrespective of its relevance (*Heed and Azañón, 2014*; *Röder et al., 2004*). The most common experimental manipulation underlying this claim is limb crossing. Crossing, say, a right arm over to the left side of space leads to different skin-based (here: right body side) and external (here: left side of space) spatial codes of a tactile stimulus delivered to the right hand. A task-irrelevant tactile stimulus delivered to a crossed right hand accelerates visual discrimination in the right visual field if it precedes the visual target stimulus by 60 ms, but on the left side if it leads by 180 ms or more (*Azañón and Soto-Faraco, 2008*). Thus, responses to visual targets were faster after anatomically congruent tactile cues (e.g., tactile stimulus on crossed right hand, visual target in right hemifield) at short cue-stimulus intervals, but after externally congruent tactile cues (e.g., tactile stimulus on the left hand crossed over to the right side, visual target in right field) at long cue-stimulus intervals. Such effects are usually interpreted as evidence that tactile remapping – the precise computation of the external tactile stimulus location – is automatic and forms the basis for the performance enhancement at this external location.

The same conclusion has also been drawn from results obtained with the tactile temporal order judgment (TOJ) task; in this task, participants report which of two successive tactile stimuli, each presented to a different body part – typically the two hands – occurred first (*Heed and Azañón, 2014*; *Shore et al., 2002*; *Yamamoto and Kitazawa, 2001*). When the time interval between the two stimuli is short, participants sometimes choose the wrong stimulus. Notably, stimulus confusion is much more prominent when the arms are in a crossed than uncrossed posture. This is surprising because the TOJ task asks about the identity of the touched limb, and, in theory, it would be irrelevant to this question where the hand was in space. That limb crossing, nevertheless, affects TOJ implies that posture cannot be strategically ignored, but is automatically incorporated into the hand assignment.

Several explanations have been put forward to account for crossing effects in tactile limb assignment. First, it has been suggested that touch location, once it is remapped, is retained only in an external-spatial code, and the original skin location is discarded in the process. To report which body part has been touched, the brain must then reversely determine which limb was located at the computed external location at the time of the touch (*Kitazawa, 2002*; *Kitazawa et al., 2008*; *Yamamoto and Kitazawa, 2001*). We refer to this suggestion as the *space-to-limb reconstruction hypothesis*. When applied to errors in the TOJ task, this hypothesis implies that participants correctly remap the two tactile stimuli into external space, but then reconstruct erroneously which hand was at the first spatial location.

A second explanation assumes that TOJ errors reflect the conflict between different codes used for stimulus location. When the limbs are crossed, skin-based and external-spatial codes point to different sides of space, and this conflict must be resolved, a process that takes time and is error-prone (*Röder et al., 2007*; *Simon et al., 1970*). In this view, the TOJ crossing effect is a marker for the presence of conflict and, thus, for the fact that remapping into an external-spatial code has taken place. Notably, the interpretation that the TOJ crossing effect derives from a remapped stimulus location is indirect because participants only report a binary decision about which hand was stimulated, not the spatial location of the perceived stimulus. Increasing the distance between the uncrossed hands can slightly reduce errors in TOJ (*Gallace and Spence, 2005*; *Roberts et al., 2003*; *Shore et al., 2005*), and the TOJ crossing effect is smaller when the hands' positions additionally differ in height or depth (*Azañón et al., 2016*). These graded modulations of the TOJ have led to the claim that the TOJ paradigm is an implicit index of tactile remapping (*Azañón et al., 2015*; *Badde and Heed, 2016*; *Heed and Azañón, 2014*). We refer to this suggestion as the *stimulus switch hypothesis*. It implies that participants have correctly remapped the two stimuli into space, but have incorrectly resolved the conflict between the different spatial codes of the first stimulus, consequently assigning the incorrect stimulus to the first time point; as a consequence, participants incorrectly report the hand that received the second stimulus.

Importantly, both hypotheses outlined above assume that touch is automatically remapped to its veridical external location. However, recent experiments have cast doubt on whether this is actually the case. For instance, if a tactile stimulus is presented during an arm movement, and participants indicate the stimulus's location by pointing to its external location after the movement, they make systematic localization errors (*Dassonville, 1995*; *Maij et al., 2011b*; *Maij et al., 2013*; *Maij et al., 2017*; *Watanabe et al., 2009*). Importantly, because these errors differ for fast and slow movements, it has been suggested that participants do not compute the precise spatial location of a stimulus when it occurs, but instead infer spatial location post-hoc by estimating hand location at the perceived time of the tactile stimulus (*Maij et al., 2017*). We refer to this suggestion as the *time reconstruction hypothesis*. Accordingly, errors in the TOJ task would occur because participants first choose the incorrect hand, and then derive stimulus location based on that hand's position at the time of the first stimulus. Note, that here participants merge the correct, first stimulus's time with the incorrect, second stimulus's hand. For the present study, the key claim of the *time reconstruction hypothesis* is that stimulus location is only computed after the hand has been chosen. This feature is at odds with the idea that tactile judgments are based on automatic spatial remapping (*Heed et al., 2015a*; *Shore et al., 2002*; *Yamamoto and Kitazawa, 2001*), according to which the stimulus location is determined first and then used to make the hand assignment – in fact, the time reconstruction hypothesis reverses the dependency between localization and limb assignment proposed by the other theoretical accounts.

Here, we assessed hand assignment and spatial localization of tactile stimuli presented during movement. Our objective was to test whether TOJ responses mark the use of the stimulus's external-spatial location constructed in response to the stimulus, or whether instead participants estimate stimulus location post-hoc by integrating the hand movement trajectory with stimulus time. In other words, we aimed to directly contrast the three discussed hypotheses for tactile localization: the space-to-limb reconstruction hypothesis, the stimulus switch hypothesis, and the time reconstruction hypothesis.

We presented human participants with two tactile stimuli during a bimanual movement and assessed which hand participants perceived to have been stimulated first (TOJ hand assignment), as well as exactly where in space participants had perceived the first stimulus (tactile stimulus localization). The experimental logic, and its relation to the three tested tactile localization hypotheses, are illustrated in *Figure 1*. Because tactile stimuli were presented shortly before, after, and during the time of movement, their spatial location depended on their timing relative to the movement. This allowed us to determine which tactile location participants had perceived when they had made a hand assignment error in the TOJ task. Contrary to common opinion, TOJ errors were not associated with the location of the second, incorrect stimulus. Instead, when participants chose the incorrect hand, they reported its location at the time point at which the first, correct stimulus had occurred. Thus, participants constructed stimulus location by combining the position of the incorrectly chosen hand with the stimulus timing that belonged to the other, non-chosen hand's stimulus, resulting in reported locations at which no stimulus had ever occurred. This finding invalidates current explanations of crossing effects as being based on the remapped external-spatial location of the tactile stimulus.

## Results

### Experiment 1

Participants adopted a start posture with their hands resting on a table and their arms stretched out in an uncrossed or crossed posture (see *Figure 1A,B* for an illustration of experimental conditions and trial timing). A tone then instructed a movement of the two hands about 30 cm toward their body, bringing the arms into either an uncrossed or crossed arm end posture (see *Figure 1A*). Shortly before, during, or shortly after the movement, participants received two tactile stimuli, one on each hand, with a stimulus onset asynchrony (SOA) of 110 ms. At this SOA, participants often misreport which of the two stimuli occurred first, both when the arms are still (*Heed and Azañón, 2014*; *Shore et al., 2002*; *Yamamoto and Kitazawa, 2001*) and during movement (*Heed et al., 2015b*; *Hermosillo et al., 2011*). After the bimanual movement, participants reported on which of the two hands the first stimulus had occurred by reaching with this hand to the perceived external location

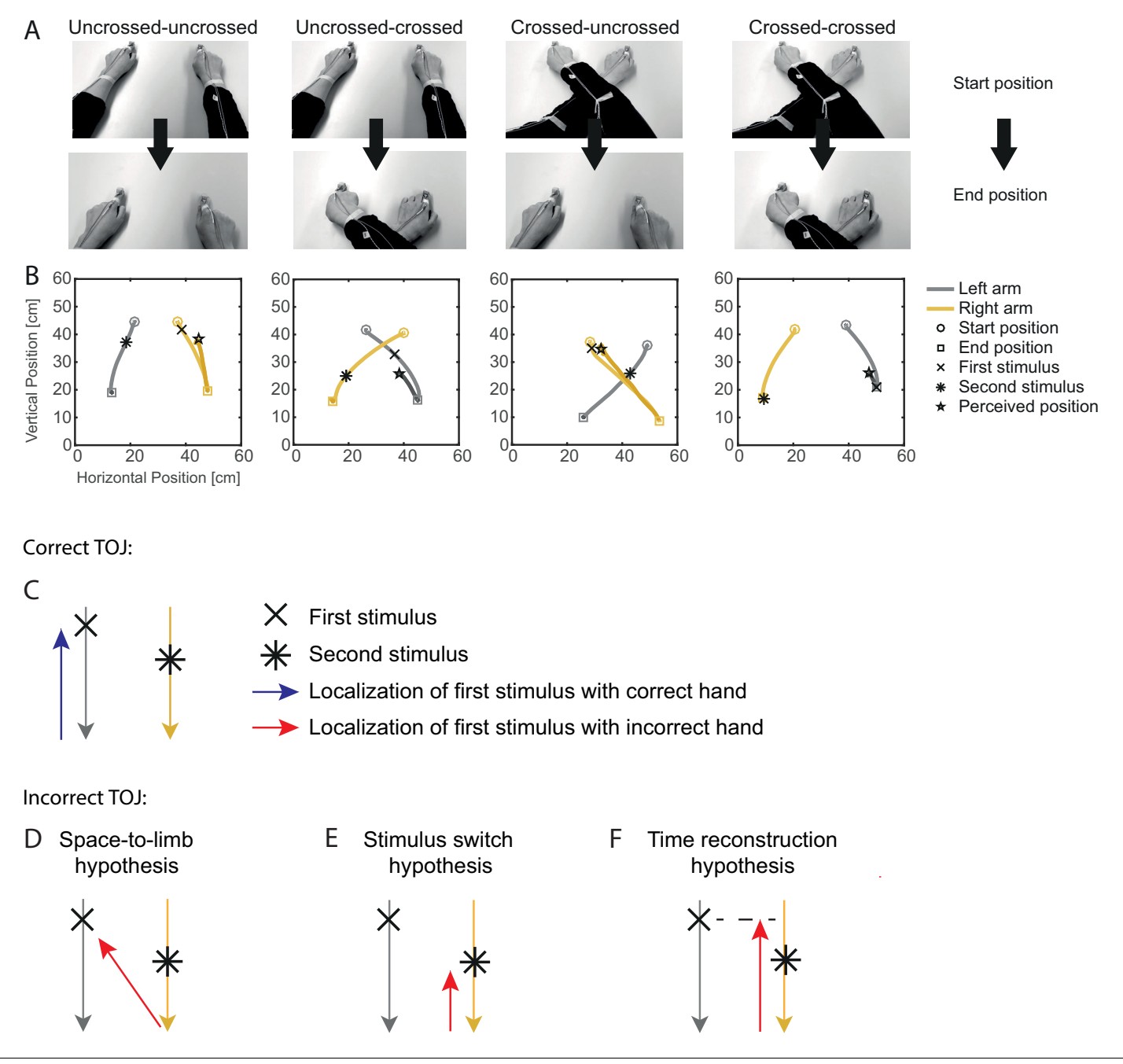

**Figure 1.** Experimental conditions of Experiment 1 and predictions of the tested tactile localization hypotheses. (A-B) Experimental procedure. (A) The arms moved from an uncrossed or crossed start posture to an uncrossed or crossed arm end posture. (B) Representative examples of TOJ trials showing the bimanual movement (gray, left hand; yellow, right hand) for the four combinations of uncrossed and crossed start and end postures, as well as the reach-to-point movement of the hand at which the first tactile stimulus was reported. (C) Illustation of a correct TOJ trial: the stimulus is assigned to the correct hand, which points to the correct location. Gray (yellow) traces illustrate the left (right) hand's movement toward the body, here during a trial from an uncrossed start to an uncrossed end posture. The blue arrow indicates the movement of the correctly assigned hand toward the location of the first stimulus (cross). (D-F) Illustration of the three hypotheses that may account for TOJ errors. The red arrows indicate the movement of the incorrectly chosen hand. (D) Space-to-limb reconstruction hypothesis: participants point with the incorrect hand at the external location of the first stimulus. (E) Stimulus switch hypothesis: participants point with the incorrect hand at the external location of the second stimulus. (F) Time reconstruction hypothesis: participants point with the incorrect hand at the location at which that hand was at the time of the first stimulus.

of the stimulus (see *Figure 1B*). The response, thus, contained two components: the hand to which the first stimulus was assigned, and explicit spatial localization of this stimulus.

## Hand assignment

In a first step, we verified that hand assignment in the TOJ task was modulated by hand crossing and timing of stimuli relative to the movement (*Heed et al., 2015b*; *Hermosillo et al., 2011*). We measured TOJ performance as the percentage of correct reports of which hand had been stimulated first in the TOJ task, as indicated by the hand that participants used for their localization response (see *Figure 2*). Stimuli could occur during all times (see Methods for details), so we binned the binary (correct/incorrect) TOJ response data into four movement phases – stimulation before movement onset, during first and second half of movement, and after movement offset – to assess the modulation of TOJ performance by stimulus time relative to the bimanual movement.

In accordance with previous findings, TOJ performance declined in the crossed compared to the uncrossed posture (*Heed and Azañón, 2014*), and depended on the posture at the time of stimulation (*Heed et al., 2015b*; *Hermosillo et al., 2011*). For instance, for the uncrossed-uncrossed movement condition (see *Figure 2*, dark-green data points), the probability of a correct response was high compared to the crossed-crossed movement condition (see *Figure 2*, light magenta data points) throughout all movement phases. For the conditions with a postural change (uncrossed-crossed, crossed-uncrossed, see *Figure 2*, dark-magenta, light-green data points) the probability of correct responses was modulated by the posture at the time of stimulation. A generalized linear mixed model (GLMM) with factors start posture, end posture, and movement phase revealed significance for all main effects and interactions (see in *Supplementary file 1*). With movement phase, the effect of start posture (see *Figure 2*, dark vs. light colors) declined, whereas the effect of end posture (see *Figure 2*, green vs. magenta colors) increased. For instance, for movements from an uncrossed to a crossed posture, TOJ performance was better during the first two movement phases, that is, when the hands were still uncrossed, than during the last two movement phases, that is, when the hands were crossed.

In sum, TOJ performance in our first experiment reflected known modulations of hand posture and movement timing. Participants made, on average, more than 15% TOJ errors even with uncrossed hands. This high error rate is due to the use of the short SOA of 110 ms (*Heed et al.,*

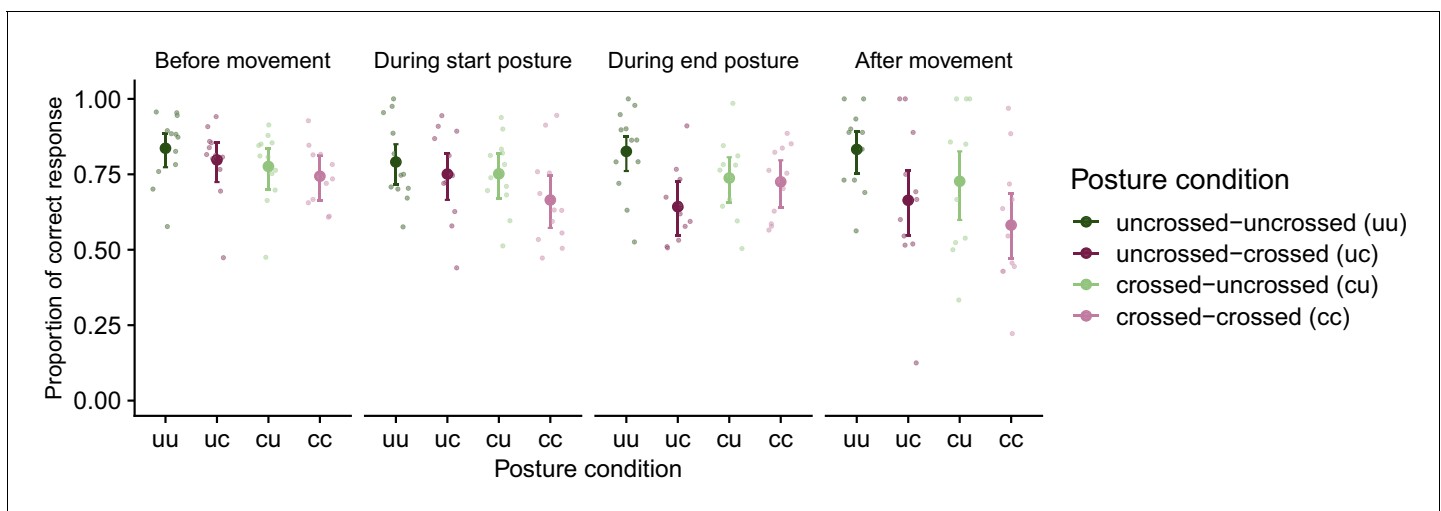

**Figure 2.** TOJ task performance in Experiment 1. Proportion of correct hand assignment across movement conditions (uncrossed-uncrossed, uncrossed-crossed, crossed-uncrossed, crossed-crossed) in the four phases of the bimanual movement (before movement, during start posture, during end posture, after movement). For conditions without a postural change (i.e., uncrossed-uncrossed, crossed-crossed), trials were assigned as 'during start posture' if the first stimulus occurred during the first temporal half of the movement and they were assigned as 'during end posture' if the first stimulus occurred during the second temporal half of the movement. For conditions with a postural change (i.e., uncrossed-crossed, crossed-uncrossed), trials were assigned as 'during start posture' if the first stimulus occurred before the postural change and as 'during end posture' if the first stimulus occurred after the postural change. Error bars denote 2 s.e. from the mean; asymmetry is due to nonlinear conversion from the GLMM's logit scale to percentage correct. Large symbols are group means, small symbols are individual participants' performance.

*2015b*; *Heed and Azañón, 2014*; *Shore et al., 2002*; *Yamamoto and Kitazawa, 2001*), and is an intended outcome of the experimental paradigm, allowing, as a next step, comparison of spatial localization responses for incorrect TOJ trials across all postures.

## Explicit tactile localization in space is unaffected by hand posture

Having verified that TOJ hand assignment showed the well-known effects of posture, we next turned to tactile stimulus localization. Localization errors are computed as the spatial difference (calculated as the signed difference in the direction along the path of the reporting hand, see Methods for details) of the perceived stimulus location and the hand's true position at stimulus presentation. From previous studies involving single stimuli and unimanual movements, it is known that participants make systematic localization errors when they retrospectively point to the spatial location of a tactile stimulus that was presented while the target limb was moving. More specifically, localization is systematically biased in the direction of the movement during the initial part of a movement, and in the opposite direction during the final part of the movement (*Dassonville, 1995*; *Maij et al., 2013*; *Maij et al., 2017*; *Watanabe et al., 2009*), resulting in systematic localization error curves with positive values indicating errors in movement direction and negative values indicating errors in the opposite direction. For short tactile stimuli, as those used here, the respective biases extend to presentation times before and after the movement (*Watanabe et al., 2009*; *Maij et al., 2013*; *Maij et al., 2017*). This pattern of movement-related directional biases was evident also in the present data (see *Figure 3AB* for an example of a single participant). Critically, bias was comparable across all four posture conditions (see *Figure 3C*).

To validate that localization behavior in our task was not biased by the specifics of the TOJ task, participants performed a simpler 1-stimulus control task in separate blocks of the experiment. While making bimanual movements with uncrossed and crossed start and end postures, they received a single tactile stimulus and pointed to it, as in the 2-stimulus task (see Methods for details). Participants virtually always indicated correctly which hand had received the stimulus (average percentage correct, 99.5%). Critically, localization error curves were indistinguishable from the task with two stimuli (see *Figure 3D*), indicating that tactile localization was affected neither by task difficulty nor by other aspects particular to the TOJ task.

## Explicit tactile localization is directed toward the assigned hand

We have so far assessed localization performance in trials in which participants had made a correct TOJ hand assignment (referred to as correct TOJ trials from hereon). We now turn to localization errors in incorrect TOJ trials. These errors allow differentiating between the three hypotheses about how participants determine stimulus localization in tactile decision paradigms (see *Figure 1D–F*).

We first turn to the space-to limb reconstruction hypothesis. It posits that tactile perception takes place in space rather than on the body; thus, a limb assignment entails computing which limb was at the first spatial location. Thus, in our task, responses with the incorrect hand would result from assigning the incorrect hand to the correct spatial location of the first tactile stimulus (see *Figure 1D*). Accordingly, the assigned, incorrect hand should be directed to the location at which the stimulus of the other, correct hand had occurred, and the reported stimulus location in incorrect TOJ trials should scatter around the movement trajectory of the correct hand. Contrary to this prediction, participants consistently pointed to locations scattered around the movement trajectory of the assigned, incorrect hand, indicating that the chosen stimulus had been perceived on the incorrect hand (see *Figure 4* for the localization responses of the same participant as shown in *Figure 3*). Thus, localization behavior did not support the implication of the *space-to-limb reconstruction hypothesis* that the correct external-spatial location is simply assigned to a wrong limb.

## Localization aims at the assigned hand's position at the time of the first tactile stimulus

Given that participants localized the stimulus along the assigned hand's trajectory, two possibilities remain as to which stimulus location was associated with erroneous responses (see *Figure 1*). The *stimulus switch hypothesis* posits that the two stimuli were localized correctly, and one is chosen for the response. In incorrect TOJ trials, participants would confuse the two stimuli and report the second stimulus by pointing at its location with the respective, incorrect hand. In this case, participants

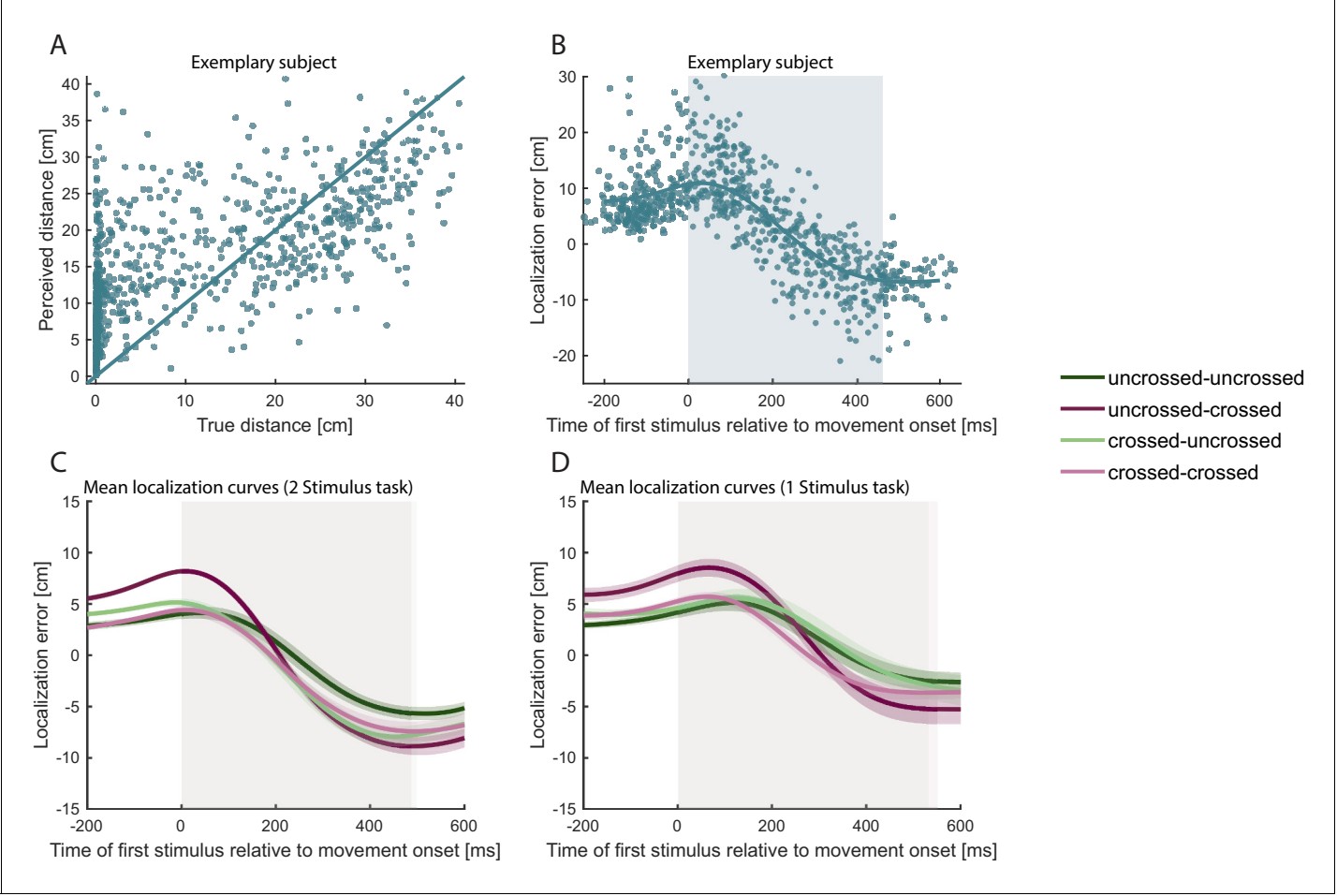

**Figure 3.** Localization errors systematically vary with the progression of the movement. (A) Distance of reported hand position from hand position at movement start ('perceived distance'; calculated as difference in the direction of the trajectory of the reporting hand) plotted against the distance of true hand position at the time of tactile stimulation from movement start ('true distance') of all correct TOJ trials of a single participant in Experiment 1. Each dot represents a single trial. The solid line represents veridical localization. (B) Mean localization error (teal line) of all correct TOJ trials of the same participant as in A. Each dot represents the localization error of a single trial, that is the difference between indicated hand position at the end of the trial and true hand position at the time of tactile stimulation. Blue shading represents the average movement time, with 0 ms = movement start. Note, that the localization error is positive at the beginning of the movement, indicating error in the direction of the movement. Localization error is negative toward the end of the movement, indicating error against the direction of movement. (C-D) The localization error pattern of the correct trials in B was evident across all participants for both the 2 stimulus experimental task (C) and for the 1 stimulus control task (D) and for all posture combinations. Traces reflect the group mean, shaded areas around the traces reflect s.e.m. The shaded regions in the background indicate the average movement duration, which differed slightly between conditions (see in *Supplementary file 1*).

should point to where the hand was positioned at the time point of the *second*, erroneously chosen stimulus (referred to as time 2 from hereon). In contrast, the *time reconstruction hypothesis* assumes that participants always use the correct, first time point (time 1 from hereon), and determine the position of the assigned response limb at this time point. This hypothesis predicts that, in incorrect TOJ trials, participants point to where the incorrectly assigned hand was positioned at the correct time, that is, time 1. Note, that no tactile stimulus occurred at this external-spatial location, because it combines the time of the first, correct stimulus with the movement trajectory of the second, incorrect stimulus's hand.

We test between the predictions of these hypotheses by comparing the localization error curves in correct and incorrect TOJ trials. In the case of correct TOJ trials, we assume that participants aimed, as instructed, at the position of the correct hand at time 1. Therefore, we derive the localization error curve as the spatial difference of perceived location and hand position at time 1 (see *Figure 5*, dark blue lines). However, we can also derive a localization error curve for correct trials under

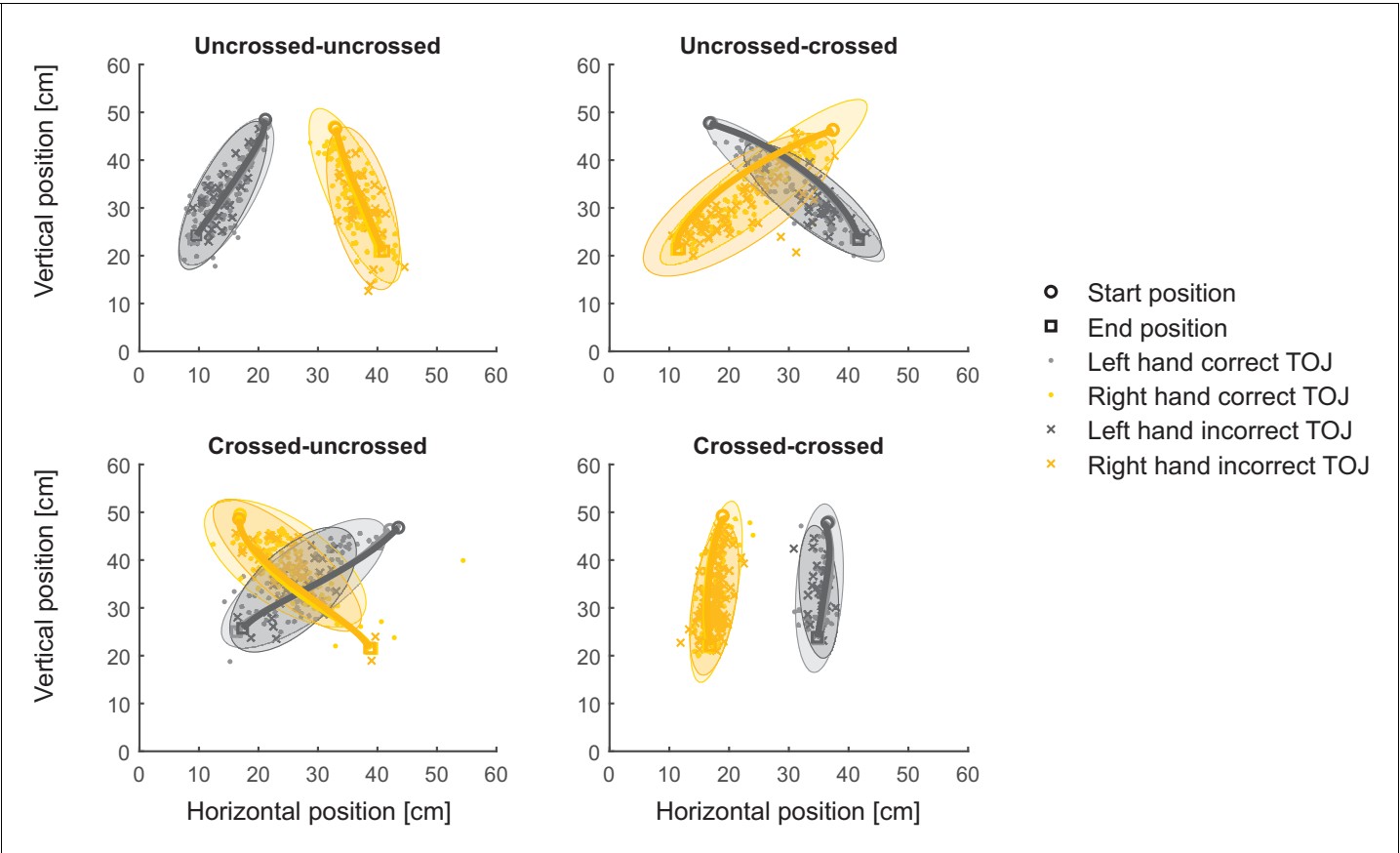

**Figure 4.** Localization responses of Experiment 1 are at odds with the space-to-limb reconstruction hypothesis. Average reach trajectories (solid lines with finger position at movement onset indicated as circles, and at movement offet as squares) and localization responses (i.e., finger positions in the horizontal plane at the end of the reach-to-point movement indicating the location where the participant perceived the first stimulus) for the different movement conditions. Data are from the same participant as in *Figure 3AB*. Ellipses represent 95% of the variability of localization responses and show large overlap for correct and incorrect TOJ trials. The space-to-limb reconstruction hypothesis would predict that, during error trials, participants point with the incorrectly assigned hand to the location of the correct stimulus; thus, if this hypothesis were correct, orange ellipses should overlay with light gray ellipses, and dark gray ellipses should overlay with yellow ellipses.

the assumption that participants pointed toward the hand's position at time 2. To derive this hypothetical curve, we calculated the spatial difference of participants' localization responses and the hand's position at time 2, rather than time 1 (see *Figure 5*, light blue lines). The time 2 error curve is shifted to the left, or 'backwards' in time, relative to the time 1 error curve. This is because, for the time 2 curve, the assumed 'true' target location is the hand's position 110 ms further into the movement, due to the SOA between the two tactile stimuli. Accordingly, each assumed target location is closer to the movement's end by the trajectory the hand has moved during the 110 ms interval between the two stimuli.

The first, time 1 error curve can now serve as a template of a localization error curve if the participant truly aimed at the hand's position at time 1. The second, time 2 error curve, in contrast, serves as a template of a localization error curve if the participants had truly aimed at the hand's position at time 2. For incorrect TOJ trials, we do not know whether participants aimed at where the incorrectly chosen hand was positioned at time 1 or at time 2. The rationale of our analysis, thus, is to compare the localization error curves of incorrect TOJ trials with the template localization error curves derived from correct TOJ trials (see *Figure 5* and Materials and methods).

Localization errors of incorrect TOJ trials overlapped with localization errors of correct TOJ trials at time 1 for each of the four start and end posture combinations. To quantify this further, we computed the temporal shift required to align localization errors of incorrect TOJ trials with those of correct TOJ trials (see *Maij et al., 2009*). If, in incorrect TOJ trials, participants aimed for the incorrect

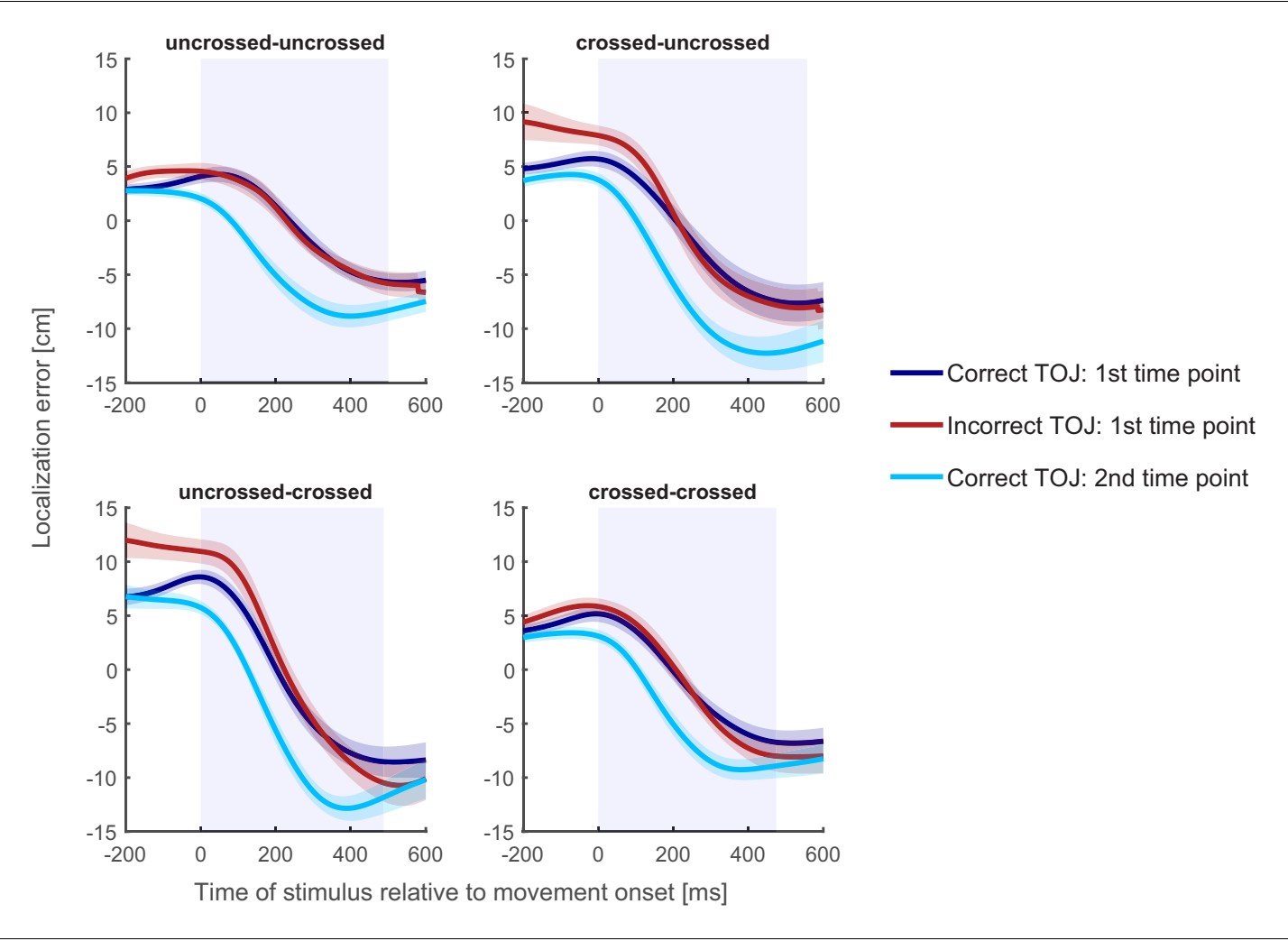

**Figure 5.** Localization curves, averaged across participants, for each of the four posture conditions in Experiment 1. Curves of incorrect TOJ trials (red) show a similar pattern as the localization curves of the correct TOJ trials at time 1 (dark blue), but not as the localization curves of the correct TOJ trials at time 2 (light blue). Traces reflect the mean localization error, shaded areas around the traces reflect s.e.m. across participants. The shaded regions in the background represent the average movement time.

hand's position at time 1, then the temporal shift should be zero relative to the localization error curve in correct TOJ trials relative to time 1; furthermore, it should be about −110 ms (negative denoting a shift toward left, see above) compared to the localization error curve of correct TOJ trials relative to time 2. If, however, participants aimed for the incorrect hand's position at time 2, the shift pattern should be exactly reversed, that is, zero compared to the second template curve, and around +110 ms compared to the first template curve.

*Figure 6* displays the temporal shift between the localization error curve of incorrect TOJ trials and the error curves at time 1 and time 2 derived from correct TOJ trials. We fitted a linear mixed model with two factors: factor Posture Condition coded the four combinations resulting from uncrossed and crossed start and end postures. Factor Reference Time Point coded whether the localization error curve for correct TOJ trials was computed relative to time 1 or time 2. The dependent variable was the time shift that best aligns the localization error curve between correct and incorrect TOJ. This analysis revealed a significant effect of Reference Time Point ($\chi^2(9,10)=15.55$, p<0.001), indicating that the time shift required to align the localization error curves of correct and incorrect TOJ trials differed depending on whether localization error of correct TOJ trials was computed based on time 1 or time 2. In contrast, there was no effect of Posture Condition ($\chi^2(7,10)=1.57$, p=0.67) or interaction ($\chi^2(7,10)=4.53$, p=0.21) between the two factors. This latter result

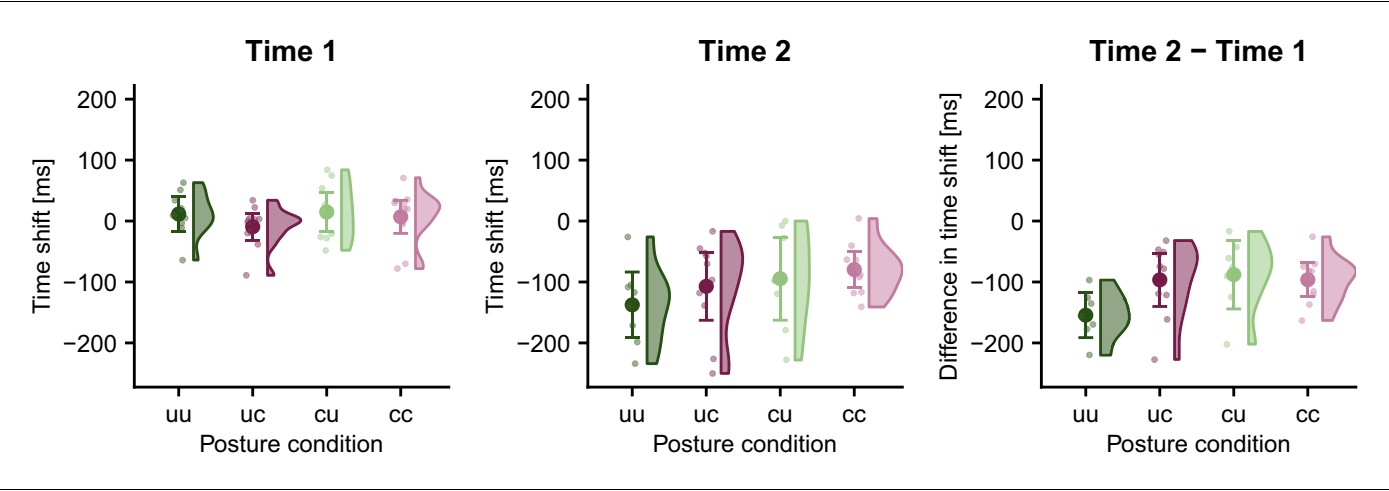

**Figure 6.** Time shift of stimulus localization error in TOJ error trials relative to time 1 (left panel) and time 2 (middle panel) for the four posture conditions in Experiment 1. The temporal shift of the localization error curve was significantly different from zero when calculated relative to time 2, but not relative to time 1; this result is consistent with the time reconstruction hypothesis, but not with the stimulus switch hypothesis. This pattern was similar across all participants as demonstrated by the differences in time shift between time 2 and time 1 (right panel). Data are visualized with raincloud plots (*Allen et al., 2019*) that display probability density estimates, condition averages (large symbols), and individual participants (small symbols). Error bars denote 95% confidence intervals.

indicates that the relationship of localization in correct and incorrect TOJ trials held across all postures; this result is also illustrated by the very similar relationship of the different localization error curves in the four panels of *Figure 5*. Thus, whereas limb posture affected hand assignment, it did not affect tactile localization.

To assess whether localization in incorrect TOJ trials aimed at a location related to time 1 or time 2, we tested the respective time shifts required to align the localization errors of the two types of trials against zero. The time shift between localization errors for correct TOJ trials at time 1 and incorrect TOJ trials, averaged across the four posture conditions, was 6 (s.e. 3) ms; if this value is not significantly different from 0, then an LMM of only this condition should not improve by inclusion of an intercept, as the latter would model the deviation of the average shift away from 0. The comparison of a model with and without intercept did not provide statistical evidence to reject a zero time shift ($\chi^2(1)=0.84$, p=0.36). Null findings are difficult to interpret in the context of frequentist statistics. Therefore, we complemented our analysis by a Bayesian analysis comparing a model with only a random participant factor with a model that, in addition, included a population intercept, equivalent to the linear mixed model reported above. The population-level intercept estimate was 6 ms and the 95% confidence range [−8.28; 19.27 ms] included 0. Model comparison via leave-one-out cross-validation found the model without intercept to be more credible than the model with the population intercept that would have been indicative of a non-zero localization error shift between correct and incorrect TOJ tials (difference of expected log predictive density (ELPD) for second as compared to first model: −0.7, s.e. 0.8; stacking weights for model-combined predictive distributions: 1, 0).

We ran the same analyses for the time shifts required to align localization error curves between correct and incorrect TOJ trials when correct trials' error curve had been calculated relative to time 2. In contrast to the results for time 1, the average time shift between localization errors for correct TOJ trials at time 2 and incorrect TOJ trials was −105 (s.e. 7) ms, and a model without intercept fit this condition significantly worse than a model with the intercept ($\chi^2(1)=19.38$, p<0.001). The significant difference to time 2 suggests that participants did not aim at the position of the second stimulus in incorrect TOJ trials.

While neither the non-significant difference to time 1 in the LMM analysis, nor the Bayesian parameter estimate including 0 statistically imply equality of the error curves in correct and incorrect TOJ trials, these statistical results are consistent with the two conditions being equal, and they suggest that, if a difference exists, it is small. Furthermore, the time shift of −105 ms for time 2 closely

matches the stimulus SOA of 110 ms, further suggesting that, in error trials, participants did not aim for hand location at the second, but rather at the first time point. Corroborating this conclusion, the Bayesian 95% interval [−138; −76 ms] of the intercept estimate includes −110 ms, and comparison of Bayesian models for time 2 with and without intercept favored the model including the intercept (difference in ELPD from first to second model, −1.4, s.e. 1.8, stacking weights: 0.017, 0.983).

## Experiment 2

The results of Experiment 1 suggest that when asked to localize the external location of the first of two tactile stimuli applied in succession to different hands, participants chose which hand received the relevant stimulus and then inferred the position of the chosen limb at the time point of the first stimulus. Consequently, when participants chose the incorrect limb, stimulus location was determined as the location at which the incorrect hand was at the correct (first) time point. While these results support the *time reconstruction hypothesis*, Experiment 1 tested only a single SOA of 110 ms between the two tactile stimuli. If our conclusions drawn from Experiment 1 are correct, then localization of stimuli assigned to the incorrect hand should always depend on the first stimulus's time, independent of SOA.

Experiment 2 tested this conjecture. Again, participants judged which hand had received the first of two tactile stimuli during a bimanual movement and then located the stimulus perceived to have occurred first. We presented tactile stimuli with four different SOAs: 60, 85, 110, and 135 ms. As explained in Experiment 1, the shift between the time 1 and time 2 curves of correct TOJ trials reflects the SOA of the two tactile stimuli. Accordingly, the two template curves are further apart the larger the SOA (compare light vs. dark blue lines in panels A-D of *Figure 7*). As the estimated localization error curves in Experiment 1 were similar for all combinations of the hands' start and end posture, Experiment 2 involved only reaches from an uncrossed to an uncrossed posture and from a crossed to a crossed posture. This strategy minimized obstruction of motion tracker markers and homogenized movement time across conditions (see in *Supplementary file 1*). Experiment 2 was conducted in a different lab than Experiment 1 and used different equipment, re-written experimental code, different experimenters, and new analysis scripts (see Material and methods for details). Moreover, given the uncertainty indicated by the ELPD standard errors in the Bayesian model comparisons of Experiment 1, we increased our sample size and acquired a higher number of trials to further scrutinize the reliability of our results.

## Hand assignment

In accordance with Experiment 1 and previous findings (*Heed et al., 2015b*; *Hermosillo et al., 2011*), TOJ performance in Experiment 2 was modulated by hand posture and SOA. At all SOAs, participants made large amounts of errors, ensuring that a sufficient number of trials were available to analyze incorrect TOJ trials. Detailed results are reported in the Supplementary Information (*Figure 7—figure supplement 1*; in *Supplementary file 1*).

## Explicit stimulus localization in space

Complementing the findings from Experiment 1 and further corroborating the *time reconstruction hypothesis*, localization errors of the incorrect TOJ trials largely overlapped with the localization errors of correct TOJ trials at time 1 for each of the four SOAs (see *Figure 7*) and for each participant and posture condition (see *Figure 7—figure supplement 2–4*).

*Figure 8* shows the temporal shift between the localization error curves of incorrect TOJ trials and the error curves of correct TOJ trials relative to time 1 and time 2 for the four SOAs.

A linear mixed model with factors SOA and Reference Time Point (localization errors for correct TOJ trials computed relative to time 1 vs. time 2) and time shift of the error curve between correct and incorrect TOJ trials as dependent variable revealed a significant main effect of Reference Time Point ($\chi^2(9,10)$=21.33, p<0.001) and a significant Reference Time Point $\times$ SOA interaction ($\chi^2(7,10)$ =9.90, p=0.02). The time shift between localization errors for correct TOJ trials at time 1 and incorrect TOJ trials, averaged across all four SOA conditions, was 8 ms. As in Experiment 1, an LMM with one common intercept for all SOAs did not fit the data better than a model without an intercept ($\chi^2(1)$=1.24, p=0.27). Similarly, allowing for individual intercepts per SOA did not improve the goodness of fit, ($\chi^2(3)$=2.74, p=0.43). Thus, none of the tested models provided statistical evidence to

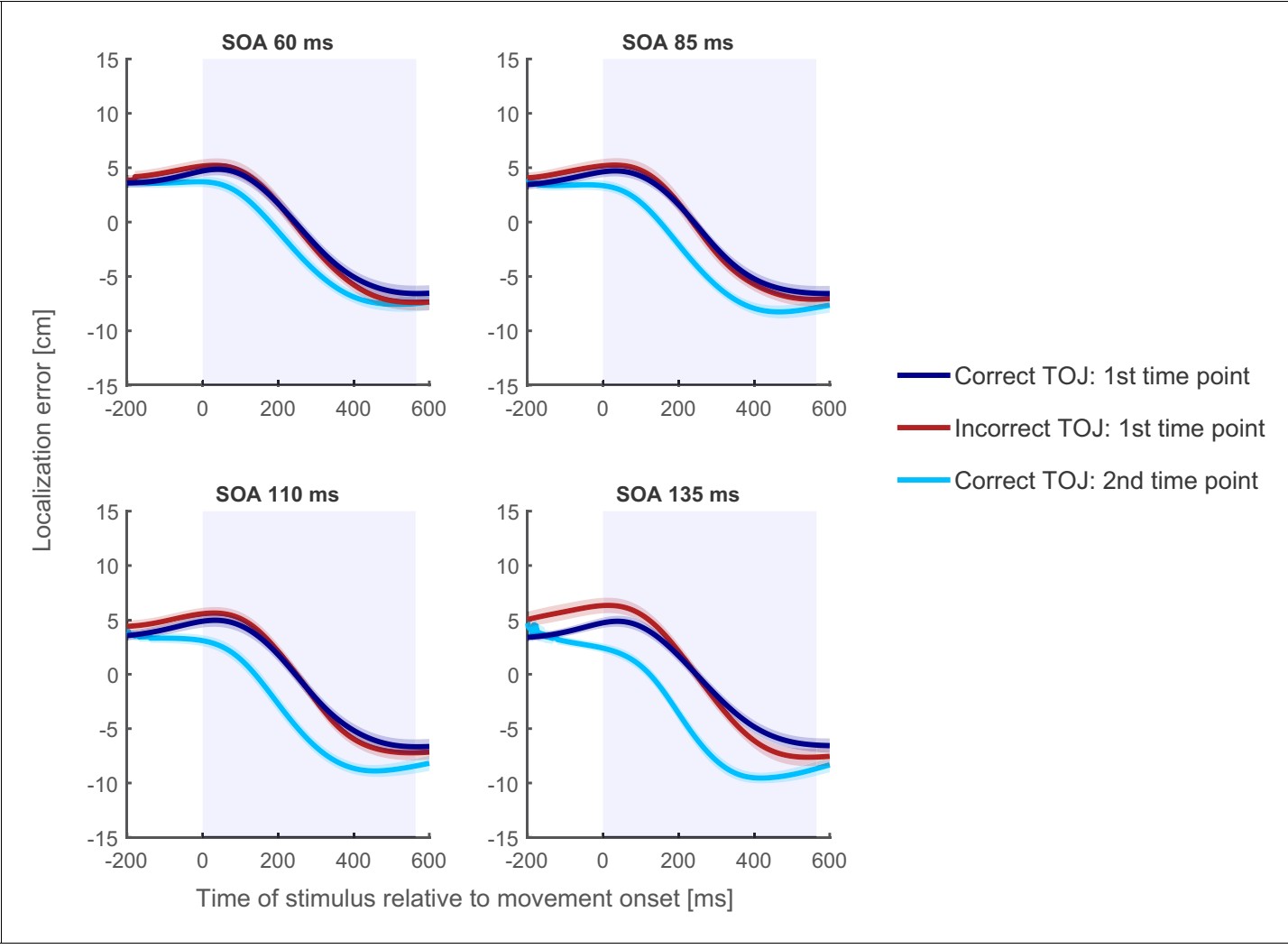

**Figure 7.** Localization curves, averaged across participants and posture, for each of the four SOAs in Experiment 2. Curves of incorrect TOJ trials (red) show a similar pattern as the localization curves of the correct TOJ trials at time 1 (dark blue), but not as the localization curves of the correct TOJ trials at time 2 (light blue). This pattern was highly similar across all participants and also when calculated separately for each posture condition (see Supplementary Information). Traces reflect the mean, shaded areas around the traces reflect s.e.m. The shaded regions in the background represent the average movement time.

The online version of this article includes the following figure supplement(s) for figure 7:

**Figure supplement 1.** TOJ performance of Experiment 2.
**Figure supplement 2.** Localization performance in the uncrossed-uncrossed condition of Experiment 2.
**Figure supplement 3.** Localization performance in the crossed-crossed condition of Experiment 2.
**Figure supplement 4.** Single participant example of localization performance in Experiment 2.

reject a zero time shift with respect to time 1. These results were again supported by parameter estimates of Bayesian models equivalent to the afore-mentioned LMMs, which estimated both an intercept across all SOAs and individual intercepts per SOA to lie in intervals that included 0 (see Supplementary Information). Model comparison via leave-one-out cross-validation found a model without population intercept to be more credible than a model with a common intercept (difference of ELPD: −0.2, s.e. 0.6) and a model with individual intercepts per SOA (ELPD: −2.0, s.e. 2.2; stacking weights for the three models: 0.911, 0, 0.089).

In contrast, the average time shifts between localization errors for correct TOJ trials at time 2 and incorrect TOJ trials were −52 (s.e. 12) ms, −75 (s.e. 12) ms, −87 (s.e. 14) ms, and −104 (s.e. 13) ms for the SOA 60, 85, 110, and 135 ms, respectively. A model with a common intercept for all SOAs explained the data significantly better than a model without an intercept ($\chi^2(1)=29.27$, p<0.001). A

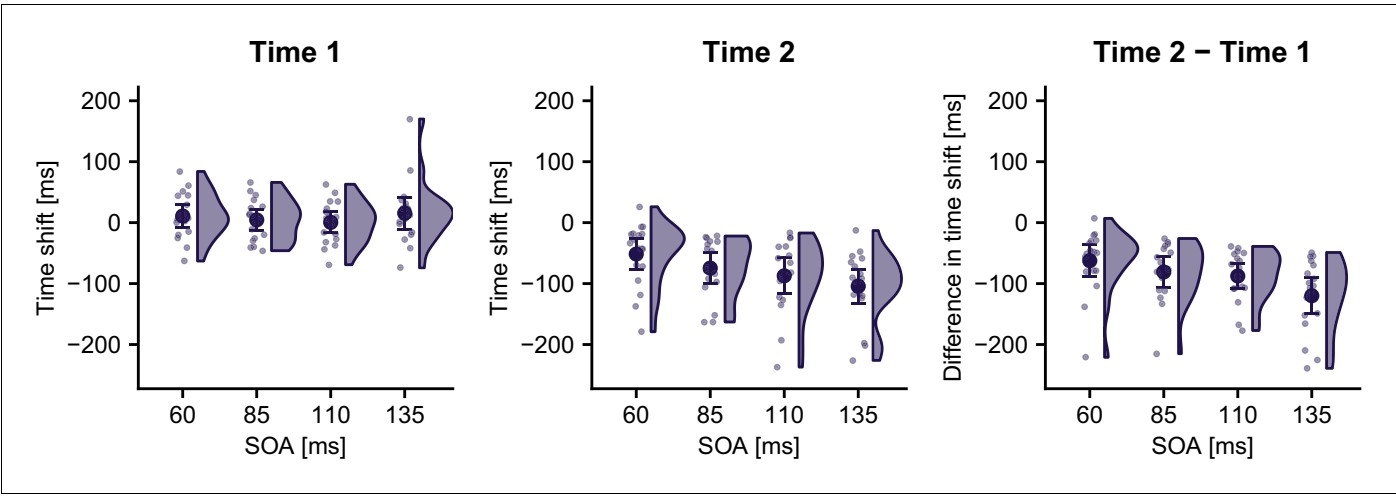

**Figure 8.** Time shift of stimulus localization error in incorrect TOJ trials relative to time 1 (left panel) and time 2 (middle panel) for the four SOAs in Experiment 2. For all SOAs, the temporal shift relative to time 1 was not significantly different from 0. In contrast, the time shift was significantly different from zero for all SOAs when calculated relative to time 2, and it was numerically similar to the respective SOA. These results are consistent with the time reconstruction hypothesis, but not with the stimulus switch hypothesis. This pattern was similar across all participants as demonstrated by the differences in time shift between time 2 and time 1 (right panel). Data are visualized with raincloud plots (*Allen et al., 2019*) displaying probability density estimates, condition averages (large symbols), and individual participants (small symbols). Error bars denote 95% confidence intervals.

model allowing for different intercepts for each SOA further improved the goodness of fit ($\chi^2(3)$ =14.3, p=0.003), indicating that localization curve's time shift relative to time 2 depended on the respective SOA (see *Supplementary file 1* for Bayesian model estimates). Model comparison via leave-one-out cross-validation found a model with individual SOA intercepts to be more credible than a model with a common intercept (difference of ELPD: −5.2, s.e. 3.8) and a model without intercept (ELPD: −7.0, s.e. 4.6; stacking weights for the three models: 0.096, 0.041, 0.863).

In sum, localization curves reflected the increase of, and shift values were numerically close to, the stimulus SOAs. Yet, the time shift value for the 110 ms SOA in Experiment 2 was smaller than that of Experiment 1 (-87 vs. −105 ms). In fact, average time shift values of Experiment 2 seemed to underestimate the true SOA in Experiment 2, although the Bayesian 95% intervals of the intercept estimates included the true SOA for all but the largest SOA (i.e., 135 ms, see *Supplementary file 1*). Furthermore, at time 1, the estimated intercepts were all slightly (albeit non-significantly) positive (*Figure 8* left panel, *Supplementary file 1*), and model comparison only slightly favored the no-intercept model. When one considers the difference in time shift (i.e., Time 2 – Time 1), the estimated values match the true SOA more closely (−63, −81, −88, and −120 ms for SOAs 60, 85, 110, and 135 ms, respectively; see *Figure 8*, right panel). We note that time shift calculations are based on a sliding Gaussian average across noisy, nonlinear patterns of localization errors, and so absolute shift values may not exactly reflect the stimulus SOAs.

## Discussion

The aim of our study was to test whether participants represent the remapped spatial location of tactile stimuli when they make spatial decisions about tactile stimuli. Participants indicated both the target limb and the perceived location in space of the first of two tactile stimuli in a tactile TOJ task. Presentation of stimuli during movement implied that stimulus location depended on stimulation time, allowing us to determine the relationship of stimulus timing and perceived stimulus location in space. If participants had first computed the spatial location, and then derived which limb had occupied this location at the time of stimulation (space-to-limb hypothesis, see *Kitazawa, 2002*), incorrect hand assignment should have been associated with external localization along the trajectory of the correct hand; we did not find evidence for such behavior. If participants had represented stimulus location and stimulated limb together, and simply confused the two due to conflict between different spatial codes (*stimulus switch hypothesis*), incorrect hand assignment should have been

associated with spatial localization at the location of the incorrect limb at the second stimulus's time. Localization error curves were incompatible with this view, as their systematic bias differed significantly from a hypothetical localization curve, derived from correct trials, relating to stimulus time 2. Instead, when participants chose the incorrect hand, their localization errors implied that they had aimed at that hand's location at the time of the first, correct stimulus, evident in a close match of localization curves of correct and erroneous TOJ trials when computed relative to stimulus time 1. In other words, participants derived the reported stimulus location by combining the time of the first, correct stimulus with the trajectory of the second, incorrectly chosen hand, effectively indicating a location at which no stimulus had occurred – consistent with the *time reconstruction hypothesis*. This behavior was evident for all combinations of uncrossed and crossed start and end postures, as well as for all tested SOAs between the two tactile stimuli. TOJ errors, thus, did not simply reflect temporal confusion of two stimuli; instead, localization in TOJ error trials marks the computation of tactile stimulus location based on correct stimulus timing and movement information of a (correctly or incorrectly) implied body part. Accordingly, limb crossing affected hand assignment, but not stimulus localization.

The pattern of hand assignment errors was in line with previous studies: Participants made more TOJ hand assignment errors in conditions that involved hand crossing than in conditions in which the hands were uncrossed (see *Figure 3*; *Heed et al., 2015b*; *Hermosillo et al., 2011*; *Shore et al., 2002*; *Yamamoto and Kitazawa, 2001*). These reliable findings support the interpretation that categorical decisions in touch, such as choosing which limb was stimulated, are affected by weighted integration of different spatial aspects of the tactile stimulus and configuration of the body (*Badde et al., 2019*; *Badde and Heed, 2016*; *Heed and Azañón, 2014*). In contrast, localization error patterns were similar across uncrossed and crossed start and end hand posture conditions, suggesting that arm posture during stimulation did not affect localization responses (see *Figures 4* and *5*). In particular, localization errors exhibited comparable spatial biases over time in uncrossed and crossed conditions. Furthermore, localization error scattered around the chosen hand was not biased toward the other hand (see *Figure 4*), an effect one might have expected if, like hand assignment, spatial localization was subject to weighted influence of the tactile stimulus's anatomical origin as coded by a body-based reference frame.

The dissociation between TOJ hand assignment and localization responses indicates that the two phenomena do not reflect the same process. It is widely assumed that the weighted integration of spatial factors reflected by tactile limb crossing effects trades off the anatomical and the external location of a tactile stimulus (*Badde and Heed, 2016*; *Cadieux and Shore, 2013*; *Kitazawa, 2002*; *Shore et al., 2002*; *Yamamoto and Kitazawa, 2001*). This assumption requires that an external location is constructed as a prerequisite for assigning a stimulus to a hand. Our present finding that participants incorrectly localize tactile stimuli associated with incorrect limb choice, in contrast, implies that not stimulus location determines hand assignment, but vice versa, hand assignment determines perceived stimulus location.

This conclusion is incompatible with the common view that crossing effects, obtained in experiments that require categorical decisions such as TOJ, are an implicit indicator of precise tactile localization and tactile remapping. This is a strong claim that may invalidate the experimental logic of numerous papers that have applied this logic. However, the present results are corroborated by another recent study that has challenged the view that errors in tactile categorical response paradigms reflect a conflict between anatomical and external-spatial coordinates. In that study, participants performed TOJ of tactile stimuli presented to uncrossed and crossed hands and feet (*Badde et al., 2019*). In each trial, two stimuli were randomly presented to two of the four limbs. In some trials, participants reported the first touch on a limb that had not been stimulated in this trial. For instance, after stimulation of the left hand and the right foot, a participant may have indicated that the first stimulus had occurred on the right hand. These TOJ errors systematically depended on different anatomical features such as the type (hand or foot) of the correct limb and its body side. Critically, neither the side of space of the limb that had received the correct stimulus, nor the spatial distance between stimulus and response limb affected TOJ errors in this task. Like the present results, these findings are incompatible with the prevailing view that crossing effects reflect conflict during the integration of anatomical and spatial stimulus location. In fact, the two studies complement each other in that we show here that stimulus location is not used for hand assignment, and

*Badde et al., 2019* suggest which information is instead used to choose between the two hands in a categorical tactile-spatial task.

Notably, other manipulations that have been used to argue for the relevance of precise spatial representations in tactile decisions can be framed in such a feature-based account as well. For instance, the TOJ crossing effect was reduced when the two hands' positions differed in height or in depth (*Azañón et al., 2016*). However, this manipulation may have simply introduced an additional, non-metric feature that helped representing the two choice options as different, thus improving TOJ choice based on a categorical feature rather than on metric distance. Similarly, whereas effects of distance between the two hands during a TOJ have been interpreted as implying a metric representation of stimulus location (*Roberts et al., 2003*; *Shore et al., 2005*), others have reported such effects only for very small (3 cm), but not other (10 cm and larger), distances between the hands (*Kim and Cruse, 2001*), again suggesting that they may reflect categorically coded spatial features, not metric stimulus location.

The reason for the apparent contradiction between previous research and these new findings stems from the fact that typical experimental designs have presented tactile stimuli to stationary hands. With such designs, one cannot disentangle whether TOJ crossing effects indicate conflict between tactile anatomical and (precise) spatial stimulus location or whether the conflict apparent in limb crossing indicates other aspects, such as the tactile features identified by *Badde et al., 2019*. Our experiments tease these possible explanations apart and suggest that limb crossing paradigms that require limb choices about the origin of touch likely reflect the integration of categorical, tactile-spatial stimulus features. We propose that automatic effects, such as crossmodal, tactile-visual cueing (*Azañón and Soto-Faraco, 2008*), too, are based on such feature-based processing. In contrast, precise stimulus location is not among the pieces of information that are integrated for automatic, tactile-spatial coding, contrary to what has regularly been implied. Instead, precise location of the tactile stimulus in space is inferred post-hoc only when required.

Furthermore, our proposal contrasts with the suggestion that TOJ errors are due to temporal confusion of the two stimuli, hypothesized to occur due to slowing of a neural clock mechanism because crossed postures induce higher cognitive load as compared to uncrossed postures (*Kitazawa et al., 2008*). This hypothesis is based on the assumption that tactile locations are represented correctly but ordered incorrectly in time. It is incompatible with our finding that participants localized touch in incorrect TOJ trials at the location of the incorrect hand at the correct, first stimulus time point. Moreover, an account based on time confusion is specific to the TOJ paradigm, in which participants compare two stimuli. In contrast, a feature-based account generalizes to other experimental paradigms, including ones that present only a single tactile stimulus (*Azañón et al., 2010*; *Azañón and Soto-Faraco, 2008*; *Badde et al., 2016*; *Badde et al., 2019*).

Our study exploited systematic localization errors when a stimulus is presented during a movement of the arm. A modulation of spatial localization by movement of the respective sensors is not unique to touch. When a brief flash is shown during a smooth pursuit or saccadic eye movement, its localization is perceived with a bias in the direction of the eye movement shortly before and during the first half of the eye movement, and in the opposite direction at the end of the eye movement (*Matin and Pearce, 1965*; reviewed by *Schlag and Schlag-Rey, 2002*). As suggested here for touch, visual mislocalization during saccades, too, depends on temporal processing. For instance, irrelevant auditory temporal information can influence the perceived location of a flash near the time of saccades and result in a temporal shift of the visual localization error curve (*Binda et al., 2010*; *Maij et al., 2009*). Moreover, when a red flash was presented around the time of a saccade on a split green/red background, participants sometimes reported that the red flash had occurred on the (same-color) red background; these reports of an objectively impossible perception (flash on same-colored background) were best explained by integration of temporal uncertainty of the flash's timing and eye position (*Maij et al., 2011a*). A computational model for these temporal-spatial phenomena faithfully replicates the observed spatial biases for both vision and touch. Illustrated for the case of tactile localization on the arm, the model assumes temporal uncertainty about tactile stimulus occurrence relative to the arm movement and combines a probability distribution of the possible stimulus time with the perceived arm movement trajectory (*Maij et al., 2013*; *Maij et al., 2017*; see *Maij et al., 2011b*, for the visual analogue). The present results, too, are compatible with the temporal uncertainty model. Independent of which hand the stimulus was assigned to, the temporal estimate of the tactile stimulus was identical, resulting in identical localization error profiles, based on

the trajectory of the chosen hand, in correct and incorrect TOJ trials. Thus, the time-based mechanism that leads to the seemingly surprising perception of spatial locations at which no stimulus really occurred may be task- and domain-general.

To summarize, we observed the typical dependence of tactile TOJ responses on limb posture, with higher error rates when the hands are crossed rather than uncrossed. Explicit localization responses of the stimulus chosen as having occurred first were incompatible with theoretical accounts that posit confusion of yoked stimulus representations that encompass the independently determined external-spatial location of tactile stimuli, or projection of a body part onto the determined spatial location of a stimulus. Instead, participants chose one hand presumably based on categorical stimulus characteristics such as the stimulated body side (*Badde et al., 2019*), and then combined the time point associated with the first stimulus with the chosen arm's trajectory. After hand assignment errors, participants, thus, effectively referenced a post-hoc constructed spatial location at which no stimulus had ever occurred.

## Materials and methods

Data for the presented analyses as well as code to run analyses and create figures are provided at the Open Science Framework website, https://osf.io/ybxn5/.

### Participants

Experiment 1 was performed at the Faculty of Psychology and Human Movement Science of the University of Hamburg. Twelve right-handed participants (aged 19–31 years, 7 female) gave informed consent to take part in the experiment. The study was part of a research program approved by the ethics committee of the German Psychological Society (DGPs). Experiment 2 was preregistered at the Open Science Framework website (https://osf.io/qyzgb). A sample size of 20 participants was defined a priori. We collected data from 20 individuals from Bielefeld University. We excluded data of 1 participant from analyses as s/he did not follow the instructions and most of the time localized the tactile stimulus at the start or end position, but not along the movement trajectory. Furthermore, we excluded data of another participant as s/he only completed 384 trials in total. As any form of data acquisition was stopped in our lab beginning of March 2020 due to the spread of the corona virus, we did not collect data from replacement participants. Our sample thus consisted of 18 participants (aged 18–25 years, 15 female). The experiment was approved by the ethics committee at Bielefeld University (Ethical Application Ref: 2017–114).

Participants provided written informed consent and were compensated with €7/hr or received course credit. All participants had normal or corrected-to-normal vision and did not have any known perceptual, motor, or neurological disorders. Participants took part only if, in a screening experiment, they exhibited a TOJ crossing effect at the SOA used in the main experiment. We used this screening procedure because individual response patterns in tactile experiments involving hand crossing are quite variable (*Badde et al., 2016*; *Cadieux et al., 2010*; *Yamamoto and Kitazawa, 2001*); however, crossing effects are highly reliable across the entire population, so that our screening procedure does not preclude generalization.

### General setup

Participants were blindfolded. They sat on a chair at a table. A tactile stimulator (Oticon BC 461-0/12, Oticon Ltd., London, UK) was attached to the phalanx media of each index finger. Stimulation consisted of 200 Hz vibration for 10 ms. To mask any noise of the vibrators, participants wore earplugs and heard white noise through speakers (Experiment 1) or wore sound-attenuating headphones (Superlux HD669, Superlux Enterprise Development, Shanghai, China; Experiment 2).

### Experiment 1

#### Apparatus, task and procedure

The position of each index finger in space was recorded with an Optotrak active, infrared marker motion tracking system (Northern Digital Inc, Waterloo, Ontario, Canada) at a sampling rate of 1000 Hz. One marker was positioned on the nail of each index finger, directly next to the tactile stimulator. A data acquisition unit (Odau; Northern Digital Inc, Waterloo, Ontario, Canada; sampling rate

1000 Hz) synchronized marker position and timing of the tactile stimuli. The experiment was controlled with Matlab (Mathworks, Natick, MA, USA), using the Psychophysics Toolbox (*Brainard, 1997*) and the Optotrak Toolbox (http://www.ecogsci.cs.uni-tuebingen.de/OptotrakToolbox/).

## 2 Stimulus task

Participants moved both hands from a position of about 40 cm away from their body toward their body to a position about 10 cm away from their body. Hand start and end posture were uncrossed and crossed, varied in blocks of 50 trials in pseudo-randomized order (see *Figure 1A*). In each trial, a tone instructed the movement start. At a random time (presented between 50 and 800 ms after the tone, drawn from a square distribution) before, during, or after the movement, two tactile stimuli were applied, one to each hand, at an SOA of 110 ms; the left-right order of stimuli was pseudo-random. Upon movement completion, participants moved the index finger that they had perceived to have been stimulated first to the location of the first stimulus on the table. The hand remained in this location until a tone, presented 2.5 s after the initial movement cue instructed them to lift the index finger; this finger lift was used to identify the response hand during trajectory analysis. Subsequently, participants repositioned the hands to their start locations. We acquired 300 trials of each posture combination. To compensate for obstruction of motion tracking markers, we acquired more trials for 2 participants in the uncrossed to crossed posture and vice versa movement conditions. The experiment took approximately 4 hours, split in two-hour sessions held on different days. Practice trials were included on each day before the experiment started until the participant had understood, and felt confident with, the task. In total we acquired 14.776 trials.

## 1 Stimulus Task

The procedure was identical to the 2 Stimulus task except that participants only received one stimulus at either hand and then indicated the perceived location with the respective index finger. Participants performed 300 trials in each posture combination split in blocks of 50 trials. In 99,5% of the trials participants used the correct arm when localizing the stimulus.

## Analysis

### Data preprocessing

Start and end of the movement were determined based on a velocity threshold of 5 cm/s. We interpolated missing motion tracking data, for instance due to obstruction when the hands passed each other or due to rotation of the hands, using splines, with the restriction that movement onset and offset could be determined. Trials were discarded when (1) missing marker data could not be adequately interpolated (1 Stimulus Task: 13%; 2 Stimulus Task: 11.9%); (2) no stimulus localization response, indicated by finger lifting, could be detected (1 Stimulus Task: 2.8%; 2 Stimulus Task: 5.4%); (3) and when participants did not perform smooth, continuous, and synchronous movements with movement duration less than 200 ms or more 1000 ms (1 Stimulus Task: 3.7%; 2 Stimulus Task: 0.2%). In total, 19% (1 Stimulus Task) and 17.5% (2 Stimulus Task) of trials were removed.

### Analysis of Temporal Order Judgments (TOJ)

We considered the TOJ to be correct when the hand used for the localization response had indeed been stimulated first.

### Localization error

We calculated the localization error, that is, the difference between the true location of the index finger at the time of stimulation and the reported location, that is, index finger pointing location just before finger lifting. Errors are reported relative to the direction of the movement as a straight line between start and end position of the hand, with positive values indicating errors in movement direction toward the end position of the hand. The localization error varies systematically with the stimulus time relative to movement onset (*Dassonville, 1995*; *Maij et al., 2013*; *Maij et al., 2017*; *Maij et al., 2011b*; *Watanabe et al., 2009*). The localization errors were converted to an estimated localization curve by averaging errors using a moving Gaussian window of 75 ms across a time window of −200 to 600 ms (step size 1 ms) with respect to movement onset. For each participant and posture condition, we calculated localization curves for correct TOJ trials relative to the onset of the

first stimulus, for correct TOJ trials relative to the onset of the second stimulus, and for incorrect TOJ trials relative to the onset of the first stimulus.

## Comparison of localization errors in correct and incorrect TOJ trials

To determine whether participants localized the stimulus relative to hand position at the first or the second stimulus timepoint, we calculated – separately for each participant and posture condition – the temporal shift that would produce the smallest deviations around a single, common localization curve of the compared conditions (1) between the incorrect TOJ localization curve and the correct TOJ localization curve relative to the first stimulus time point and (2) between the incorrect TOJ localization curve and the correct TOJ localization curve relative to the second stimulus time point. Specifically, we shifted the data points of the incorrect TOJ localization curves in time from –300 to 300 ms in steps of 1 ms and calculated the squared localization error differences with an localization curve calculated from data points of the two compared conditions using a common moving Gaussian average for each time shift. We refer to the time value that minimized the summed squared error differences with this overall construction curve as time shift (*Maij et al., 2009*; *Maij et al., 2017*).

We assessed the time shift for both stimulus times and, accordingly, obtained two shift values per participant and posture condition. In some instances, we were unable to construct a time shift due to a low number of incorrect TOJ trials for that specific condition (18 cases out of 4 postures x 2 stimulus times x 12 participants = 96); these data points were treated as missing data in the linear mixed model analysis.

## Statistical analysis

We assessed statistical significance of the reported results using (Generalized) Linear Mixed Models ([G]LMM) (*Bolker et al., 2009*) as implemented in R version 3.6.1 (*R Development Core Team, 2014*) using packages lme4, version 1.1–21 (*Bates et al., 2015*), and afex version 0.26–0 (*Singmann, 2015*). We estimated intercept parameters of Bayesian mixed factorial models equivalent to LMM using packages brms, version 2.12.0 (*Bürkner, 2017*; *Bürkner, 2018*), and loo, version 2.2.0 (*Vehtari et al., 2017a*; *Vehtari et al., 2017b*).

GLMM are adequate for analysis of binary variables such as correct vs. incorrect responses in our TOJ task (*Jaeger, 2008*). Furthermore, (G)LMM are robust against missing data and account for differences in trial numbers across conditions, as present in our data. All reported statistics were computed using type 3 sums of squares, as implemented in afex. For the random structure of LMM and GLMM for TOJ analysis, we included only random intercepts, because models did not reliably converge when random slopes were included. Models that tested time shifts against zero used only the data corresponding to one particular reference time point of correct trials (localization error relative to time 1 or time 2); given that posture did not significantly modulate time shift, the respective models excluded this factor. Accordingly, we compared a model without intercept [shift ~0 + (1) | participant] against a model with intercept [shift ~1 + (1) | participant], effectively testing whether a nonzero intercept significantly improved the time shift fit.

The brms R package uses STAN as backend. We ran LMM to estimate the 95% interval of the intercepts in the different time shift models. We compared Bayesian models using the loo_compare () and stacking_weights() functions of the loo R package. The former function uses leave-one-out cross-validation to compare models by assessing the models' predictive density when each data point is omitted from fitting, whereas the latter determines the proportion with which each model's predicitve distribution should be included in an overall prediction to best account for the empirical data (*Vehtari et al., 2017b*).

## Experiment 2

### Apparatus, task and procedure

Kinematic data of the fingers were recorded using an optical motion capture system (Visualeyez II VZ4000v, Phoenix Technologies Inc, Vancouver, BC, Canada) at 250 Hz sampling frequency with markers placed on the nail of the two index fingers. The experiment was controlled with Matlab (The MathWorks Version R2015a; Natick, MA, USA) using the Psychophysics Toolbox (*Brainard, 1997*). Stimulus presentation was controlled via custom-made hardware and triggered through a digital acquisition card (PCI-6509, National Instruments, Austin, USA).

The procedure was largely similar to Experiment 1, except that (1) participants made only reaches from an uncrossed to an uncrossed posture or from a crossed to a crossed posture and (2) tactile stimuli were separated by SOAs of 60, 85, 110, or 135 ms. Posture was varied in blocks of 64 trials in a pseudo-randomized order. Within each block, SOA and which hand was stimulated first were pseudo-randomized. Participants performed 28 blocks (14 of each posture combination) for a total of 1792 trials. To compensate for marker obstruction and failure to reliably detect finger lifting, two participants performed 10 additional blocks (5 of each posture combination). As we stopped any form of data acquisition in our lab for an indefinite time period in the beginning of March 2020, 2 participants performed only 19 and 23 blocks, respectively. The experiment took about 5–6 hr to complete, split in two-hour sessions held on different days. Practice trials were included prior to each experimental session.

## Analysis

### Data preprocessing

We used custom-written Matlab scripts for processing of kinematic data. We first interpolated missing data points and resampled the data to 1000 Hz using splines, and low-pass filtered the data using a second-order butterworth filter with a cut-off frequency of 6 Hz. We determined movement onset/offset of each hand as the time of the sample in which the resultant velocity of the respective finger marker exceeded/dropped below 5 cm/s. We excluded trials when missing marker data could not be adequately interpolated (8%), when no stimulus localization response, indicated by finger lifting, could be detected (4.2%), and when participants did not perform smooth, continuous, and synchronous movements (6.8%). In total we removed 18.2% of the trials.

### Analysis of TOJ and localization error

TOJ and localization errors were determined as in Experiment 1. Because localization error curves were similar regardless of start and end posture in Experiment 1, we collapsed across postures in Experiment 2 to calculate the localization error curves; we calculated individual curves for each participant and SOA (60, 85, 110, and 135 ms). Localization error curves calculated separately for Experiment 2's two posture conditions yielded qualitatively similar results (see Supplementary Information).

### Comparison of localization errors in correct and incorrect TOJ trials

Time shift values were calculated as in Experiment 1 (separately for each participant and SOA) using a shifting window of −300 to 300 ms (step size 1 ms).

### Statistical analysis

We assessed TOJ performance using a generalized mixed model (GLMM) with factors Posture (uncrossed-uncrossed, crossed-crossed), and SOA (60, 85, 110, 135 ms). The analysis approach for the dependence of time shifts on SOAs followed a similar logic as Experiment 1. We first assessed the significance of main effects and interaction of the experimental design with afex. We then assessed whether time shifts were 0 relative to time 1 and time 2. To this end, we compared models with a random participant factor but no fixed factors and intercept [shift ~0 + (1) | participant], with a common intercept for all SOAs [shift ~1 + (1) | participant], and with individual intercepts per SOA [shift ~SOA + (1) | participant] separately for shift values relative to time 1 and time 2, respectively.

## Acknowledgements

We thank Maie Stein, Franziska Rudzik, and Nina Held for help with data acquisition; Volker Franz, Rainer Schäfer, and Renate Kiesewalter for their help with the Optotrak and ODAU setup; Conrad Alting for help with programming of Experiment 2; and Mathieu GM Koppen for help with LMM analysis. FM was supported by the Netherlands Organization for Scientific Research (NWO Veni Grant 451-12-009). TH was supported by an Emmy Noether grant of the German Research Foundation (DFG) (He 6368/1–1). WPM received support from the European Research Council (EU-ERC-283567) and the Netherlands Organization for Scientific Research (NWO-VICI: 453-11-001). We acknowledge support for the publication costs by the Deutsche Forschungsgemeinschaft and the

Open Access Publication Fund of Bielefeld University. Data and code for the present paper are available at the Open Science Framework website, https://osf.io/ybxn5/.

## Additional information

### Funding

| Funder | Grant reference number | Author |
| --- | --- | --- |
| Nederlandse Organisatie voor Wetenschappelijk Onderzoek | NWO-VENI: 451-12-009 | Femke Maij |
| Deutsche Forschungsgemeinschaft | He 6368/1-1 | Tobias Heed |
| Nederlandse Organisatie voor Wetenschappelijk Onderzoek | NWO-VICI: 453-11-001 | W Pieter Medendorp |
| H2020 European Research Council | EU-ERC-283567 | W Pieter Medendorp |

The funders had no role in study design, data collection and interpretation, or the decision to submit the work for publication.

### Author contributions

Femke Maij, Conceptualization, Data curation, Software, Formal analysis, Validation, Investigation, Visualization, Methodology, Writing - original draft, Project administration, Writing - review and editing; Christian Seegelke, Conceptualization, Data curation, Software, Formal analysis, Validation, Investigation, Visualization, Writing - original draft, Project administration, Writing - review and editing; W Pieter Medendorp, Conceptualization, Supervision, Funding acquisition, Validation, Writing - original draft, Writing - review and editing; Tobias Heed, Conceptualization, Resources, Formal analysis, Supervision, Funding acquisition, Validation, Investigation, Visualization, Methodology, Writing - original draft, Writing - review and editing

### Author ORCIDs

Christian Seegelke (iD) https://orcid.org/0000-0001-9624-6395
W Pieter Medendorp (iD) http://orcid.org/0000-0001-9615-4220
Tobias Heed (iD) https://orcid.org/0000-0001-5632-6091

### Ethics

Human subjects: The study was part of a research program approved by the ethics committee of the German Psychological Society (DGPs), advisory opinions TB 10_2011 and TB_10_2011_Add 082013. Experiment 2, which was run at a different university after the move of the last author, was again approved by Bielefeld University's ethics committee, ref.nr. 2017-114.

### Decision letter and Author response

Decision letter https://doi.org/10.7554/eLife.57804.sa1
Author response https://doi.org/10.7554/eLife.57804.sa2

## Additional files

### Supplementary files

• Supplementary file 1. Five tables reporting detailed statistical results of the two experiments (Tables 1-5).

• Transparent reporting form

#### Data availability

All data generated or analysed during this study are included in the manuscript and supporting files. Source data and Matlab/R analysis files have been provided for all data and data figures at the Open Science Framework (https://osf.io/ybxn5/).

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
