## [Decision Letter]

**Acceptance summary:**

People have a harder time reporting which of their hands was touched when their hands are crossed than when they are uncrossed. This finding has led previous studies to suggest that hand assignment depends on touch location in external space. By using an elegant bimanual reaching task, this paper reveals that the opposite is true. Touch location depends on hand assignment and not vice versa.

**Decision letter after peer review:**

Thank you for submitting your article "External location of touch is constructed post-hoc based on limb choice" for consideration by *eLife*. Your article has been reviewed by three peer reviewers, including Andrew Pruszynski as the Reviewing Editor and Reviewer #1, and the evaluation has been overseen by Tamar Makin as the Senior Editor.

The reviewers have discussed their comments with one another and the Reviewing Editor has drafted this decision to help you prepare a revised submission.

The editors have judged that your manuscript is of interest. But, as described below, additional experiments are required before it is published. We would therefore like to draw your attention to changes in our revision policy that we have made in response to COVID-19 (https://elifesciences.org/articles/57162). First, because many researchers have temporarily lost access to the labs, we will give authors as much time as they need to submit revised manuscripts. We are also offering, if you choose, to post the manuscript to bioRxiv (if it is not already there) along with this decision letter and a formal designation that the manuscript is "in revision at *eLife*". Please let us know if you would like to pursue this option. (If your work is more suitable for medRxiv, you will need to post the preprint yourself, as the mechanisms for us to do so are still in development.)

Summary:

The authors examined performance on a temporal order judgement task where participants have to indicate the location where a hand was touched by using the hand that was touched first. Participants frequently used the finger that was touched second and indicated a position on that finger's trajectory. The authors interpret this as evidence for a hypothesis that perception is constructed on demand (time-reconstruction). All the reviewers were very enthusiastic about this study and agreed that it is a very interesting and potentially important finding that could make a strong theoretical contribution to understanding how we represent the body in space. The reviewers all agreed however that there was critical data required to permit the strong claim being made. They also provide a number of additional constructive suggestions about how to improve the paper.

Essential revisions:

1) The reviewers were concerned about the generalizability of the findings. This was seen as a critical problem because the importance of the paper is directly tied to how strongly it invalidates the logic of previous papers on this topic, and the reviewers were wary of upending all this previous work on the basis of a single paradigm that could create biases due to the nature of the task. All the reviewers agreed that resolving this concern would require additional experimental work as described below.

In the interesting task designed by the authors, the participant needs to make a response with the hand they've identified as being stimulated first and then use that hand to localize where that first tactile stimulus was presented. This task could lead to the participant attempting to track the positions of both hands which could lead to them attending less to the temporal order of the two stimuli than would be expected in a typical temporal order judgement (TOJ) task. This brings two potential issues. First, crossing effects in TOJ tasks can change based on the nature of the task. For example, Roberts and Humphreys, 2008 found that if you provide participants with a TOJ task in which they respond to which frequency (high or low) or which stimulus duration (long or short) was presented first, the typical crossing effect is substantially diminished. In this task, participants may be attending more to the location of the two stimuli, and less towards discriminating the two stimuli in time and space. Although I do not have a mechanism for why it would change performance, it is possible that the nature of the task could (as seen in Roberts and Humphreys, 2008). Second, and more to the point in terms of key experimental work, would the results hold if the participant responded in some way other than making a movement with the stimulated hand? For example, by making a saccade to where they perceived the first stimulus and then verbally stating whether it was the left or right hand? Let's say that participants are paying more attention to tracking hand position, and less to the temporal order of the two stimuli as they would be in a typical, static TOJ task. When uncertain about which hand is first, the participant then simply chooses a hand and then retraces the movement of that selected hand. It is possible that having participants respond with the stimulated hand biases responses such that they are less likely to be "remapping" responses and are more errors in temporal processing.

2) It is important to provide a better link to a similar mechanism in visual perception during fast eye movements as previously described by one of the authors. Maij et al., 2011a, asked participants to indicate the location of red and green flashes presented at a red-and-green background (always on a contrasting background color). Their results were similar to the present ones: participants frequently indicated that they saw green flashes at the green part of the background (or red flashed on the red part). Most relevant to the present paper, their participants used hand movements to indicate the visually perceived location during the eye movement so related to the experimental suggestion in Essential revision #1 above. It would be useful to discuss temporal uncertainty, which seems very relevant to this paper: the reporting of an event that is a reconstruction of something that never could have been perceived and has also been used to explain the flash-lag effect.

3) The authors observe error rates during four phases of the movement (see Figure 3). This provides interesting information regarding the relationship between error rates and time – but not space. If there were some spatial remapping, one might predict that there would be increased errors as a function of the distance of the two hands. That is, as the perceived position of the hands (when the stimuli were presented were closer), errors would increase. This would apply primarily to the uncrossed-crossed and crossed-uncrossed blocks. Was this the case?

4) Some key steps in the analysis are unclear. The authors don't show the time-course of the movement and don't show whether there is a correlation between the position of the hand at a certain time and the indicated position. One would have expected that participants wouldn't make a localization error if the hand was static, e.g. before t=0 or after t=500ms and have some transient errors in between (as in Maij et al., 2011b, Figure 2). However, the errors in the present paper are largest if the hand is static during stimulation. Graphs as in Figure 5A of the present paper could in principle be the result of participants always tapping halfway the start and the end of a 20 cm movement trajectory. Why are there systematic errors when a finger is stimulated that is stationary at the start or the end? This is not accidental, it is a very consistent finding, and should be explained.

5) There is no ethics approval indicated for Experiment 2.

[Editors' note: further revisions were suggested prior to acceptance, as described below.]

Thank you for submitting your article "External location of touch is constructed post-hoc based on limb choice" for consideration by *eLife*. Your article has been reviewed by three peer reviewers, including Andrew Pruszynski as the Reviewing Editor and Reviewer #1, and the evaluation has been overseen by Tamar Makin as the Senior Editor.

The reviewers have discussed the reviews with one another and the Reviewing Editor has drafted this decision to help you prepare a revised submission.

Summary:

The reviewers were generally very satisfied with your previous responses but there exist some issues that all reviewers agree need to be addressed and re-reviewed prior to a final decision.

Revisions:

1) I am generally happy with the revised manuscript, except for the answer to comment 4. I agree with the authors that it is unlikely that addressing this issue will change the overall conclusion of the paper. The first reason is that the peculiarities are also present in the new experiment without TOJ, and the second reason is that at least in some conditions, the size of the localization errors at movement onset and offset seem too small to be compatible with not localizing at all. However, the mislocalisation pattern they report is peculiar. They write that it is common that the largest localization errors occur before movement onset and after offset. In Figure 2 of the cited paper that was mentioned in this comment clearly shows no localization errors at motion onset. Indeed, the Dassonville paper cited by the authors in their reply shows a small positive error at motion onset, but *no* error after motion offset. The pattern of errors in Dassonville is clearly different from that in the present paper (it starts and ends with zero error). I only ask to add data showing the correlation between the movement progression and the response, as was done both in the cited Maij et al., 2011 paper as well as in the Dassonville paper (in slightly different formats). Now data on the movement kinematics are missing the paper.

2) The authors argue that the data of Figure 4 support their interpretation. The elongated distribution of data points indeed suggests that the participants not always moved to the same position, but to a (random?) position near the center segment of the hand's path. It would have been nice if this figure would have been for the same participant as Figure 3, and the average path of the hands would have been added. The authors did not follow my suggestion to do so because they are afraid that readers would misinterpret the errors as the hand-paths are variable. I would be surprised if the lateral variability of the hand-paths would be larger than the lateral variability in the localizations. I hope the authors will add the average path; in any case, they could add the average starting points and endpoints of the movements (with 95% confidence ellipses).

---

## [Author Response]

Essential revisions:1) The reviewers were concerned about the generalizability of the findings. This was seen as a critical problem because the importance of the paper is directly tied to how strongly it invalidates the logic of previous papers on this topic, and the reviewers were wary of upending all this previous work on the basis of a single paradigm that could create biases due to the nature of the task. All the reviewers agreed that resolving this concern would require additional experimental work as described below.In the interesting task designed by the authors, the participant needs to make a response with the hand they've identified as being stimulated first and then use that hand to localize where that first tactile stimulus was presented. This task could lead to the participant attempting to track the positions of both hands which could lead to them attending less to the temporal order of the two stimuli than would be expected in a typical temporal order judgement (TOJ) task. This brings two potential issues. First, crossing effects in TOJ tasks can change based on the nature of the task. For example, Roberts and Humphreys, 2008, found that if you provide participants with a TOJ task in which they respond to which frequency (high or low) or which stimulus duration (long or short) was presented first, the typical crossing effect is substantially diminished. In this task, participants may be attending more to the location of the two stimuli, and less towards discriminating the two stimuli in time and space. Although I do not have a mechanism for why it would change performance, it is possible that the nature of the task could (as seen in Roberts and Humphreys, 2008). Second, and more to the point in terms of key experimental work, would the results hold if the participant responded in some way other than making a movement with the stimulated hand? For example, by making a saccade to where they perceived the first stimulus and then verbally stating whether it was the left or right hand? Let's say that participants are paying more attention to tracking hand position, and less to the temporal order of the two stimuli as they would be in a typical, static TOJ task. When uncertain about which hand is first, the participant then simply chooses a hand and then retraces the movement of that selected hand. It is possible that having participants respond with the stimulated hand biases responses such that they are less likely to be "remapping" responses and are more errors in temporal processing.

This comment can be summarized by two main questions: (i) does the nature of the task affect localization response, and (ii) are the results dependent on the response mode (i.e. localizing using a hand movement)? Our answer to both questions is no, which we will justify with new data, previous findings, and theoretical arguments.

Regarding the nature of our task (question 1): The comment implies that hand choice somehow “works differently” in our present experiment, thereby referring to work by Roberts and Humphreys, 2008. Indeed, these authors reported that a crossing effect was not evident when participants judged the temporal order of non-spatial characteristics of the stimulus (e.g., vibration frequency), independent of the hand at which that stimulus had occurred. In other words, there was no limb assignment of the stimulus in that experiment! In contrast, the crossing effect was evident in their experiment, as known from several previous studies, when participants had to report the hand stimulated first (i.e., make a limb assignment, as done in our study). We have shown that crossing effects occur for other limb assignment tasks than the TOJ task (Badde, Heed and Röder, 2015; Badde, Röder and Heed, 2019). Furthermore, we and others have demonstrated that increased processing load modulates, but does not abolish, crossing effects when participants make TOJ limb assignment responses (Badde, Röder and Heed, 2015). Finally, several studies have acquired TOJ responses during arm movements and have shown that TOJ hand assignment is directly linked to the posture induced by planned and/or executed arm movements (Hermosillo, Ritterband-Rosenbaum, and van Donkelaar, 2011; Heed, Möller and Röder, 2015). The critical findings of those studies are that planned or executed hand crossing impairs TOJ hand assignment performance and that planned or executed *un*crossing improves TOJ hand assignment performance. We find exactly the same performance pattern in our present TOJ task. According to previous theory, these TOJ error patterns (of previous and of the present study) would have to be due to an integration of a remapped, spatial stimulus location.

In stark contrast, localization error – that is, the difference between the stimulus location indicated by the participant vs. the true stimulus location at time of stimulation – is unaffected by the TOJ task.

We demonstrate this with new data we have added to the paper (see new Figure 3C and subsection “Explicit tactile localization in space is unaffected by hand posture”). Our participants not only performed localization in the context of a TOJ task, but also in the context of a single tactile stimulation, but we had omitted these data from the original manuscript. The single stimulus task is both a different type of task (no comparison of stimuli required) and is overall easier (i.e., it controls for task difficulty). A comparison of localization in the one-stimulus and the two-stimulus (TOJ) tasks renders virtually identical localization error curves (compare new Figure 3B vs. 3C). This finding demonstrates that localization performance in our study does not depend on the TOJ task: whether participants are handling a single stimulus or making a comparative judgment about two stimuli, their localization responses are similar for all combinations of uncrossed/crossed start and end postures. (see new Figure 3B).

Thus, TOJ were affected by hand crossing, whereas localization responses were not. This suggests that localization responses, assessed via pointing, are a trustworthy measure of tactile localization. Even if, as raised by the reviewers comment, TOJ effects were modulated by the task, this would – according to our newly added data – not affect localization responses and, more importantly, could not explain why participants pointed towards the first time point’s stimulus location in incorrect TOJ trials. We have reordered and rewritten our Results and added new data and figures to better explain the reasoning of our study and conclusions. We hope that these revisions and the addition of new data, now documented in the revised manuscript, resolve the reviewers’ concern.

Regarding the response mode (question 2): Crossing effects have been demonstrated across multiple response modes, such as saccades (Yamamoto and Kitazawa, 2001; Hermosillo, Ritterband-Rosenbaum and van Donkelaar, 2011), hand reaches (Heed, Möller and Röder, 2015), and button presses (most other papers cited in the manuscript). A direct comparison, with comparable results for saccades and button presses, was reported in Yamamoto and Kitazawa, 2001. An indirect comparison between saccades and hand reaches is possible by comparing Overvliet, Azañón and Soto-Faraco, 2011 and Brandes and Heed, 2015. . Again, responses are highly similar. Thus, one would not expect that our results would be different if another reporting method were used. Moreover, please note that participants were blindfolded in our experiments, and an experiment involving saccades to two arms on the table (i.e., on a horizontal plane in 3D, rather than on a monitor) would not only be highly technically challenging, but also induce unwanted changes to our paradigm, such as being able to see the hands, which would likely affect localization errors and hand assignment.

2) It is important to provide a better link to a similar mechanism in visual perception during fast eye movements as previously described by one of the authors. Maij et al., 2011a, asked participants to indicate the location of red and green flashes presented at a red-and-green background (always on a contrasting background color). Their results were similar to the present ones: participants frequently indicated that they saw green flashes at the green part of the background (or red flashed on the red part). Most relevant to the present paper, their participants used hand movements to indicate the visually perceived location during the eye movement so related to the experimental suggestion in Essential revision #1 above. It would be useful to discuss temporal uncertainty, which seems very relevant to this paper: the reporting of an event that is a reconstruction of something that never could have been perceived and has also been used to explain the flash-lag effect.

Thank you for this suggestion. We have added a discussion of the relationship of tactile and visual time-related localization errors to the Discussion section and now also mention that the temporal processing seen in the present study may be a domain-general mechanism.

3) The authors observe error rates during four phases of the movement (see Figure 3). This provides interesting information regarding the relationship between error rates and time – but not space. If there were some spatial remapping, one might predict that there would be increased errors as a function of the distance of the two hands. That is, as the perceived position of the hands (when the stimuli were presented were closer), errors would increase. This would apply primarily to the uncrossed-crossed and crossed-uncrossed blocks. Was this the case?

We first emphasize that the focus of our paper is not on the TOJ paradigm. The TOJ task is merely a tool to produce a sufficient number of incorrect hand assignment responses, which are required to compare localization after correct and incorrect hand choice. Any far-near effects would, therefore, be a side aspect that distracts from the paper’s key topic. Furthermore, there are important theoretical and experimental reasons to refrain from applying such an analysis applied to the present data.

Experimentally, using a static paradigm (i.e. in which the hands don’t move), previous studies showed that the TOJ is affected by the distance between the hands. We discuss the respective papers in our Introduction as well as in the Discussion, including their explanation of these results.

However, applying this type of analysis of “close vs. far” to the present experimental data, as suggested by the reviewers, suffers from important confounds that invalidate this approach in the present dynamic paradigm. First, “close” locations would always occur midway during the movement, whereas “far” postures would always occur at the beginning and end of the movement, so that the two conditions have consistent differences unrelated to the investigated question of distance. Second, the hands would always be “close” when they are shortly before or after a crossing/uncrossing, whereas the hands would usually be far apart when posture does not change during the movement (i.e., in uncrossed-uncrossed or crossed-crossed conditions). At hand distance 0 (i.e., when the hands are above each other), the participant would be unable to determine whether the hands are crossed or uncrossed, resulting in maximal uncertainty not only about spatial location, but also about a categorical representation of hand posture (uncrossed/crossed). Our recent work (Badde, Röder and Heed, 2019) has suggested that categorical information of posture is relevant information for TOJ.

Note further that the effects that were previously reported for a distance manipulation were very small compared to the crossing effects that are usually investigated with the tactile TOJ paradigm. As a side note, one of us (TH) was unable to replicate this effect in several attempts, even after contacting one of the original authors to identify unreported aspects of the setup that may have affected the results (Schicke, 2008). We also ran a distance manipulation in a TOJ paradigm across all four limbs (Badde, Röder and Heed, 2019), and it did not affect TOJ responses.

For these reasons, we prefer to refrain from testing the distance hypothesis suggested by the reviewers. We ask the reviewers to consider again the evidence we have discussed under comment 1, namely that localization behavior was independent of the different posture conditions, indicating that the two phenomena reflect different processes.

4) Some key steps in the analysis are unclear. The authors don't show the time-course of the movement and don't show whether there is a correlation between the position of the hand at a certain time and the indicated position. One would have expected that participants wouldn't make a localization error if the hand was static, e.g. before t=0 or after t=500ms and have some transient errors in between (as in Maij et al., 2011b, Figure 2). However, the errors in the present paper are largest if the hand is static during stimulation. Graphs as in Figure 5A of the present paper could in principle be the result of participants always tapping halfway the start and the end of a 20 cm movement trajectory. Why are there systematic errors when a finger is stimulated that is stationary at the start or the end? This is not accidental, it is a very consistent finding, and should be explained.

Indeed, localisation errors occur already before movement onset and also after movement offset. This is a common finding in the context of localization during movement. Participants make mistakes in the localisation of a single short stimulus near the time of the movement; this is true for saccades (e.g., Matin and Pearce, 1965) and arm movements (e.g., Dassonville, 1965). In fact, errors are often largest around movement onset and offset, as is the case also in our own data. Therefore, the localization error patterns in our study conform with previous findings. The key new finding, however, is that participants localise the relative to the first stimulation time point even when the stimulus was assigned to the incorrect hand – as if they perceived the first stimulus at that hand. We explain the mislocalization findings (including multiple references) in the beginning of the revised Results section.

The possibility that participants always simply tap midway between movement start and end point is a valid argument, and we inspected the data of each participant to ascertain this was not what they did. Please refer to our Figure 4; it illustrates localization performance of a single participant. It is clear that endpoints are not simply halfway along the arm movement.

5) There is no ethics approval indicated for Experiment 2.

Ethics information was previously present in our manuscript. We have now more clearly indicated that Experiment 2 was approved by the ethics committee at Bielefeld University (Ethical Application Ref: 2017-114).

[Editors' note: further revisions were suggested prior to acceptance, as described below.]

Summary:The reviewers were generally very satisfied with your previous responses but there exist some issues that all reviewers agree need to be addressed and re-reviewed prior to a final decision.

For final preparation of publishing our data at OSF, we re-ran all Bayesian models to be able to provide saved model objects in the repository. Due to the probabilistic nature of Bayesian sampling, the results of the Bayesian statistics have changed slightly for all models, and we have included the new values to be consistent with the models provided on OSF. Furthermore, we have added additional model comparison information in the Results.

Revisions:1) I am generally happy with the revised manuscript, except for the answer to comment 4. I agree with the authors that it is unlikely that addressing this issue will change the overall conclusion of the paper. The first reason is that the peculiarities are also present in the new experiment without TOJ, and the second reason is that at least in some conditions, the size of the localization errors at movement onset and offset seem too small to be compatible with not localizing at all. However, the mislocalisation pattern they report is peculiar. They write that it is common that the largest localization errors occur before movement onset and after offset. In Figure 2 of the cited paper that was mentioned in this comment clearly shows no localization errors at motion onset. Indeed, the Dassonville paper cited by the authors in their reply shows a small positive error at motion onset, but no error after motion offset. The pattern of errors in Dassonville is clearly different from that in the present paper (it starts and ends with zero error). I only ask to add data showing the correlation between the movement progression and the response, as was done both in the cited Maij et al., 2011 paper as well as in the Dassonville paper (in slightly different formats). Now data on the movement kinematics are missing the paper.

The localization error pattern we observed in the present study resembles the patterns reported in two other recent studies authored by two of us (Maij, Wing and Medendorp, 2013; Maij, Wing and Medendorp, 2017). The results of these three studies differ from those of the two other studies (Maij et al., 2011; Dassonville, 1995). We commented on these differences between studies in (Maij et al., 2011): the different studies had different movement speeds and stimulus characteristics, and we speculate that these differences cause the differences in the localization curves. We would like to emphasize that 1) the shape of the localization error curves is consistent across participants and the two experiments in our study as well as two previous papers and 2) the main result on which our paper is based – namely, the shift of the curves between conditions – does not depend on the particular shape of the curves.

Furthermore, we have modified Figure 3 to show the relationship between the distance of indicated hand position at the time of tactile stimulation from hand position at movement start (i.e., perceived distance) and the distance of actual hand position at the time of tactile stimulation from hand position at movement start (i.e., true distance; see Figure 3A), as was done in Maij et al., 2011 in a similar way (Figure 2A of that paper). It is evident that there is a strong correlation between true and reported stimulus location in space. This new plot, thus, complements the data shown in Figure 3B and Figure 4 and further shows that our data is incompatible with the view that participants simply pointed to the middle of their reach trajectory or any other single location, such as the start or end point of their movements. Moreover, Figure 3A further nicely illustrates the change of localization bias around the midpoint of the trajectory, with points along the first half of the movement exhibiting a positive localization error, indicated by data points falling above the diagonal line. In contrast, during the second part of the movement most data points are below the diagonal line, indicative of a negative localization error.

2) The authors argue that the data of Figure 4 support their interpretation. The elongated distribution of data points indeed suggests that the participants not always moved to the same position, but to a (random?) position near the center segment of the hand's path. It would have been nice if this figure would have been for the same participant as Figure 3, and the average path of the hands would have been added. The authors did not follow my suggestion to do so because they are afraid that readers would misinterpret the errors as the hand-paths are variable. I would be surprised if the lateral variability of the hand-paths would be larger than the lateral variability in the localizations. I hope the authors will add the average path; in any case, they could add the average starting points and endpoints of the movements (with 95% confidence ellipses).

The single subject data shown in Figure 3 and 4 were already from the same participant. We have now indicated this in the figure caption. We have also added the average reach trajectories separately for each hand on correct and incorrect TOJ trials for each panel of Figure 4.

**References**

Brandes, J., and Heed, T. (2015). Reach Trajectories Characterize Tactile Localization for Sensorimotor Decision Making. The Journal of Neuroscience, 35(40), 13648–13658. https://doi.org/10.1523/JNEUROSCI.1873-14.2015

Overvliet, K. E., Azañón, E., and Soto-Faraco, S. (2011). Somatosensory saccades reveal the timing of tactile spatial remapping. Neuropsychologia, 49(11), 3046–3052. https://doi.org/10.1016/j.neuropsychologia.2011.07.005

Schicke T (2008), Multisensory integration in peripersonal space: beyond the hands, Hamburg, University, Dissertation,

https://katalogplus.sub.uni-hamburg.de/vufind/Record/58710208X?rank=1